# Adenosine receptor 2a agonists target mouse CD11c+T-bet+ B cells in infection and autoimmunity

Russell C. Levack [1,4], Krista L. Newell [1], Berenice Cabrera-Martinez[1], Justin Cox [1], Andras Perl[2], Sheldon I. Bastacky[3] & Gary M. Winslow [1✉]

CD11c+T-bet+ B cells are recognized as an important component of humoral immunity and autoimmunity. These cells can be distinguished from other B cells by their higher expression of the adenosine receptor 2a. Here we address whether $A_{2A}$ receptor activation can affect CD11c+T-bet+ B cells. We show that administration of the $A_{2A}$ receptor agonist CGS-21680 depletes established CD11c+T-bet+ B cells in ehrlichial-infected mice, in a B cell-intrinsic manner. Agonist treatment similarly depletes CD11c+T-bet+ B cells and CD138+ B cells and reduces anti-nuclear antibodies in lupus-prone mice. Agonist treatment is also associated with reduced kidney pathology and lymphadenopathy. Moreover, $A_{2A}$ receptor stimulation depletes pathogenic lymphocytes and ameliorates disease even after disease onset, high-lighting the therapeutic potential of this treatment. This study suggests that targeting the adenosine signaling pathway may provide a method for the treatment of lupus and other autoimmune diseases mediated by T-bet+ B cells.

[1] Department of Microbiology and Immunology, Upstate Medical University, Syracuse, NY 13210, USA. [2] Department of Medicine, Division of Rheumatology, Upstate Medical University, Syracuse, NY 13210, USA. [3] Department of Pathology, University of Pittsburgh School of Medicine, Pittsburgh, PA 15213, USA. [4]Present address: University of Pittsburgh School of Medicine, Pittsburgh, PA 15213, USA. ✉email: winslowg@upstate.edu

CD11c+T-bet+ B cells, also known as Age-associated B Cells (ABCs), are a subset of B cells involved in both protective and pathogenic immune responses. CD11c+T-bet+ B cells develop in infections characterized by type I immunity, such as malaria, HIV, influenza, and hepatitis C[1–4]. CD11c+T-bet+ B cells generated in these infections are driven by Toll-like receptor (TLR) signals, and by CD40L, IFN-γ, and IL-21 produced by T-bet+CD4+ T cells[3,5–8]. CD11c+T-bet+ B cells can function as memory cells that are capable of self-renewal and that differentiate following secondary challenge[9]. Moreover, elimination of CD11c+ B cells following *Ehrlichia muris* infection ablated the switched-antibody recall-response[10]. CD11c+T-bet+ B cells are involved in the immune response to many other diseases. During malarial infection, CD11c+T-bet+ B cells generate pathogen-specific antibodies and are associated with reduced parasite-density[11–13]. CD11c+T-bet+ B cells generated during influenza infection account for 20–30% of the antigen-specific B cells[3]. In HIV infection, the CD11c+T-bet+ B cell population contains more HIV-specific B cells than classical memory B cells[2]. In addition to their function during infection, CD11c+T-bet+ B cells are also involved in pathogenic immune responses in diseases such as rheumatoid arthritis (RA), Sjøgren's Syndrome, and systemic lupus erythematosus (SLE)[14–17].

T-bet+ B cells are key mediators of disease etiology in SLE, a relapsing autoimmune disease that affects approximately 5 million people worldwide. SLE commonly manifests as facial rash, fatigue, joint pain, and disease flares, followed in some cases by end-stage renal disease and organ failure[18]. CD11c+T-bet+ B cells contribute to SLE pathogenesis, as the cells correlate with autoantibody titers and disease severity[16,19–23]. CD11c+T-bet+ B cells are responsive to TLR7 stimulation, and exhibit reduced expression of the negative proinflammatory regulators NFKBIA, TNFAIP3, TRAF5, and TRAF4, which likely contributes to their generation during SLE[16]. It is also known that CD11c+T-bet+ B cells in humans and mice can give rise to antibody-secreting cells (ASCs), and it has been proposed that SLE-derived CD11c+T-bet+ B cells are poised to differentiate into plasmablasts (PBs)[16]. Given their propensity to generate ASCs, CD11c+T-bet+ B cells likely contribute to SLE disease pathogenesis via their differentiation into ASCs and the production of autoreactive antibodies[24–27]. CD11c+T-bet+ B cells may also contribute to SLE pathogenesis via the activation of CD4+ T cells, as these B cells can also function as potent antigen presenting cells[28]. CD4+ T follicular helper cells (T_FH), in turn, can drive the formation of autoreactive germinal center (GC) B cells and ASCs, which can give rise to autoantibodies and immune complexes, ultimately leading to systemic inflammation and glomerulonephritis[29–31]. Previous work has clearly established that T-bet+ B cells can contribute to disease pathogenesis in SLE, as elimination of T-bet expression in B cells substantially reduced disease burden in lupus-prone mice[26]. Another study similarly demonstrated that depletion of CD11c+ B cells reduced anti-Smith antibodies in mice immunized with a TLR7 agonist[25]. These studies highlight the importance of CD11c+T-bet+ B cells in the pathogenesis of SLE, and suggest that targeted depletion of these cells will reduce disease severity in autoimmune diseases wherein these cells have been implicated; however, no such pharmacological treatments currently exist.

As part of studies of CD11c+T-bet+ B cells elicited by *E. muris* infection, Winslow et al. identified a number of genes that were highly expressed, relative to CD11c-negative B cells. Among these was the gene encoding the G-protein-coupled-receptor A_{2A} receptor, *Adora2a*, which was detected at levels 10-fold higher in CD11c+T-bet+ B cells, compared to CD11c-negative B cells[32]. The A_{2A} receptor is one of several receptors for extracellular adenosine and is expressed on most leukocytes, as well as neurons and endothelial cells[33–35]. Extracellular adenosine can be produced via the catabolism of ATP and NAD+ by ectoenzymes such

as CD73, CD39, and CD38, all of which are highly expressed on CD11c+T-bet+ B cells[10,36]. Adenosine signaling via the A_{2A} receptor can be immunosuppressive, by increasing regulatory T cell (T_reg) generation, inhibiting effector T cells (T_eff) and T_FH proliferation, and blocking the formation of GC B cells[37–40]. Given its immunomodulatory ability, the A_{2A} receptor has been widely studied for possible anti-tumor effects, as some tumors are known to generate large quantities of extracellular adenosine[41–43]. Adenosine production by these tumors is thought to impair anti-tumor responses by inducing anergy among T_eff cells, and by increasing the generation of T_reg cells and their production of the immunosuppressive cytokines TGFβ and IL-10[44]. Consistently, A_{2A} receptor-deficient mice showed improved tumor rejection compared to A_{2A} receptor-competent controls, although these mice were prone to develop autoimmunity[35]. A_{2A} receptor deficiency also exacerbated severe experimental auto-immune encephalitis in mice, the result of increased macrophage and lymphocyte activation and increased IFN-γ secretion[45].

The A_{2A} receptor has been implicated in other autoimmune diseases such as RA[40,46–48]. Lymphocytes from RA patients have been shown to express high levels of A_{2A} receptor, and in vitro stimulation of this receptor inhibited the production of TNF, IL-1β, and IL-6 by these lymphocytes[47,48]. A_{2A} receptor agonism has been shown to impair the production of autoantibodies and reduce disease severity[40]. In lupus-prone mice, treatment with the A_{2A} receptor agonist CGS-21680 significantly improved kidney function, although the mechanism whereby this occurred was not known[49]. Other studies have demonstrated that the A_{2A} receptor is correlated with disease severity in SLE patients, and that stimulation of this receptor in vitro reduces the production of proinflammatory cytokines by lymphocytes[50]. These studies highlight the importance of A_{2A} receptor signaling during auto-immunity and suggest that pharmacological targeting of this receptor may aid in the treatment of diverse diseases.

In the present study, we address the effect of adenosine signaling on T-bet+ B cells. We demonstrate that administration of the A_{2A} receptor agonist CGS-21680, can deplete CD11c+T-bet+ B cells in both *E. muris*-infected and lupus-prone mice. Moreover, A_{2A} receptor agonist treatment of MRL/lpr mice significantly reduces disease severity, compared to vehicle-treated controls when administered either early or late relative to the onset of disease. Our work thus describes a targeted pharmacological approach for the elimination of pathogenic CD11c+T-bet+ B cells and highlights the therapeutic potential of A_{2A} receptor agonists in the treatment of SLE, and possibly other autoimmune diseases.

## Results

**A_{2A} receptor stimulation depleted CD11c+T-bet+ B cells.** It has previously been demonstrated that the gene encoding the A_{2A} receptor is expressed at least ten-fold higher in *E. muris*-elicited CD11c+T-bet^hi B cells, relative to CD11c-negative B cells[32]. While A_{2A} receptor signaling is known to be immunomodulatory, its effects on the maintenance of CD11c+T-bet^hi memory B cells had not been addressed. Therefore, we treated *E. muris*-infected mice with CGS-21680 once daily from days 30–37 post-infection, after the CD11c+T-bet^hi B cell population is fully developed[10]. CD11c+ B cells and T-bet^hi B cells are referred to interchangeably as it has been shown that these cells represent the same population at this time point in *E. muris* infection[5]. CGS-21680 treatment eliminated nearly all the CD11c+T-bet^hi B cells in *E. muris*-infected mice within 7 days post-administration (Fig. 1a and Supplementary Fig. 1A). Bacterial load was unchanged in CGS-21680-treated mice, likely because multiple immune mechanisms (e.g., IgM) are protective in infected mice (Supplementary Fig. 1B). This B cell

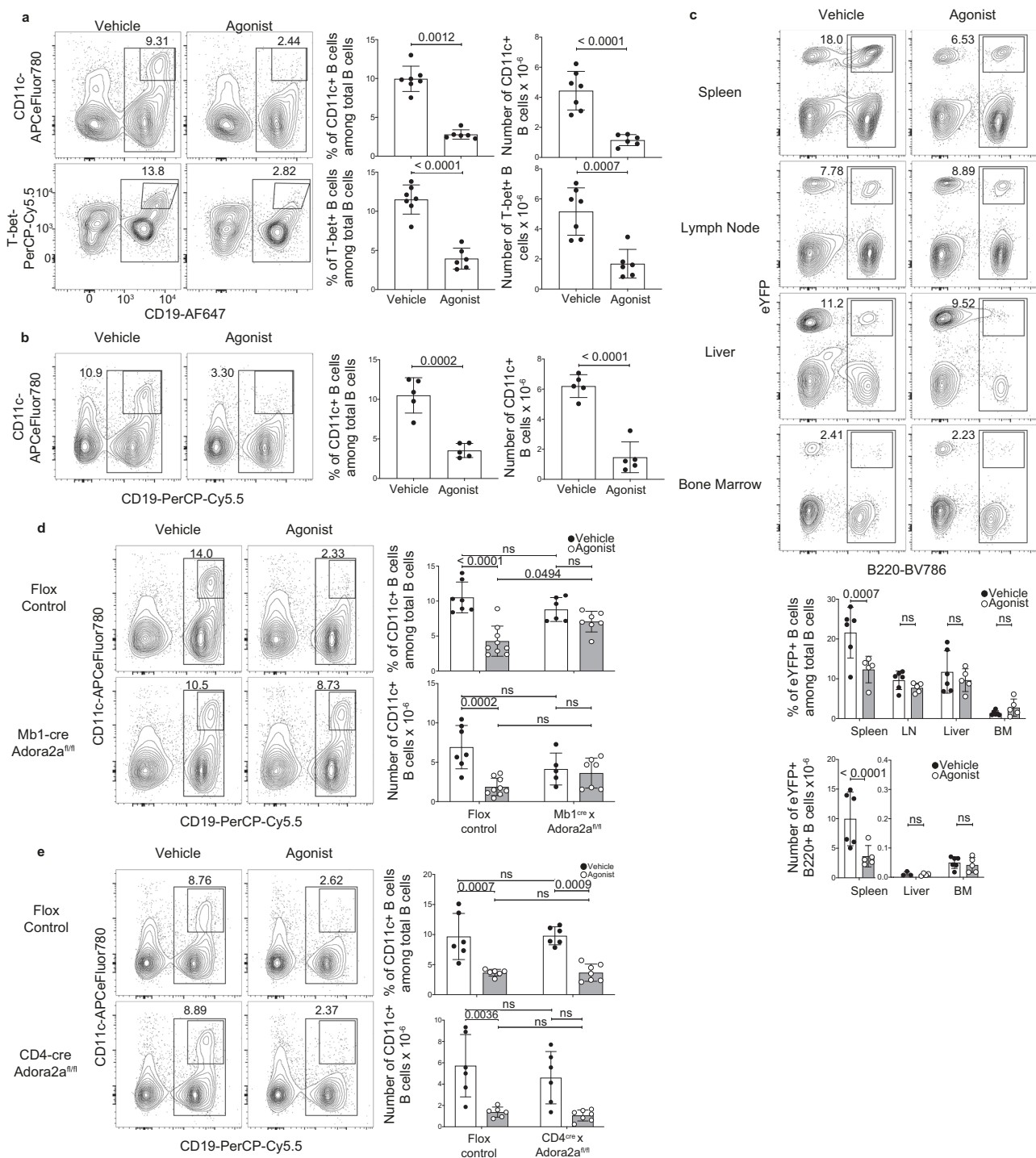

depletion occurred as early as 24 h, and a single administration of CGS-21680 was sufficient to deplete the CD11c+T-bet$^{hi}$ B cells (Fig. 1b). While $A_{2A}$ receptor stimulation depleted CD11c+T-bet$^{hi}$ B cells, CD11c-negative T-bet$^{lo}$ B cells as well as ASCs and total B cells were largely unaffected, suggesting that these cells do not express, or express less $A_{2A}$ receptor, possibly indicating a correlation between $A_{2A}$ receptor expression and T-bet expression (Supplementary Fig. 1C, D). To confirm that $A_{2A}$ receptor agonism can deplete CD11c+T-bet$^{hi}$ B cells, we treated E. muris-infected mice with an alternate $A_{2A}$ receptor agonist, Regadenoson, every other day from days 30–37 post-infection. Regadenoson treatment resulted in a decrease in CD11c+T-bet$^{hi}$ B cells in E. muris-infected mice, indicating that other $A_{2A}$ receptor agonists can also deplete

CD11c+T-bet$^{hi}$ B cells, although possibly to a lesser extent than with CGS-21680 (Supplementary Fig. 1E). We next addressed the effects of $A_{2A}$ receptor antagonism on CD11c+T-bet$^{hi}$ B cells by treating E. muris-infected mice with the $A_{2A}$ receptor antagonist Istradefylline every other day from days 30–37 post-infection. In contrast to agonist treatment, there was no impact on the number of CD11c+T-bet$^{hi}$ B cells following treatment with Istradefylline (Supplementary Fig. 1F). These data indicate that $A_{2A}$ receptor agonists can potently deplete CD11c+T-bet$^{hi}$ B cells.

Although our data suggested that CGS-21680 induced cell death in CD11c+T-bet$^{hi}$ B cells, it was formally possible that the cells downregulated expression of CD11c and T-bet, or migrated to other tissues following CGS-21680 administration. As such, we utilized

**Fig. 1 A$_{2A}$ receptor stimulation depleted CD11c$^+$T-bet$^+$ B cells. a** *E. muris*-infected female C57BL6/J mice were treated with DMSO (vehicle; $n = 7$) or CGS-21680 (agonist; $n = 6$) every day for 7 days starting on day 30 post-infection; splenocytes were analyzed by flow cytometry on day 37 post-infection. Contour plots (left) and graphs (right) show the percentages and numbers of CD11c$^+$CD19$^+$ cells (percentage: $p < 0.0012$; number: $p < 0.0001$) and T-bet$^{hi}$CD19$^+$ cells (percentage: $p < 0.0001$; number: $p = 0.0007$). Graphs represent aggregate data from two experiments. Statistical significance was determined using two-tailed *unpaired t*-tests except in the top left graph where a two tailed Mann–Whitney test was used. The p values are indicated in the graphs. **b** *E. muris*-infected female wild-type mice were treated with vehicle ($n = 5$ each) on day 30 post-infection, and splenocytes were analyzed 24 h later. Contour plots and graphs show the percentages and numbers of CD11c$^+$CD19$^+$ cells. Statistical significance was determined using two-tailed *unpaired t*-tests (percentage: $p = 0.0002$; number: $p < 0.0001$). **c** *E. muris*-infected T-bet-cre$^{ERT2}$ x Rosa26$^{eYFP}$ mice were fed tamoxifen chow from day 15–25 post-infection, followed by administration of vehicle ($n = 3, 6$) or agonist ($n = 4, 5$) every other day for 7 days, starting on day 30 post-infection; tissues were analyzed on day 37 post-infection. Contour plots and graphs show the percentages and numbers of eYFP$^+$B220$^+$ cells. Statistical significance was determined using an ordinary two-way ANOVA with Sidak's multiple comparisons test (df = 28). **d** *E. muris*-infected Adora2a$^{flox/flox}$ mice and Mb1$^{cre/+}$ x Adora2a$^{flox/flox}$ mice were treated with vehicle (flox control: $n = 7$; Mb1$^{cre/+}$ x Adora2a$^{flox/flox}$: $n = 6$) or agonist (flox control: $n = 9$; Mb1$^{cre/+}$ x Adora2a$^{flox/flox}$: $n = 7$) every other day for 7 days starting on day 30 post-infection; splenocytes were analyzed on day 37 post-infection. Contour plots and graphs show the percentages and the numbers of CD11c$^+$CD19$^+$ cells. Graphs represent aggregate data from three experiments. Statistical significance was determined using an ordinary two-way ANOVA with Sidak's multiple comparisons test (df = 25) (ns = not significant). **e** *E. muris*-infected Adora2a$^{flox/flox}$ mice and CD4$^{cre/+}$ x Adora2a$^{flox/flox}$ mice were treated with vehicle (flox control: $n = 6$; CD4$^{cre/+}$ x Adora2a$^{flox/flox}$: $n = 6$) or agonist ($n = 7$), as in **d**. Contour plots and graphs show the percentages and numbers of CD11c$^+$CD19$^+$ cells. Graphs represent aggregate data from four experiments. Statistical significance was determined using an ordinary two-way ANOVA with Sidak's multiple comparisons test (df = 21). Columns and error bars throughout indicate the arithmetic mean and SD.

T-bet-cre$^{ERT2}$ x Rosa26$^{eYFP}$ mice, wherein T-bet-expressing cells can be irreversibly marked by eYFP expression following tamoxifen administration. The mice were fed chow containing tamoxifen from days 15–25 postinfection and were then treated with CGS-21680, every other day, from days 30–37 post-infection. A similar labeling technique using AID-cre$^{ERT2}$ x Rosa26$^{eYFP}$ mice, has been used and has shown that tamoxifen-labeled eYFP$^+$ cells present after day 30 post-infection are canonical ABCs. Likewise, the majority of eYFP$^+$ cells from the T-bet-cre$^{ERT2}$ x Rosa26$^{eYFP}$ mice were CD138-negative, similar to T-bet$^{hi}$ B cells typically seen at this time point in wild-type *E. muris*-infected mice (Supplementary Fig. 2A). We observed a significant reduction of eYFP$^+$ B cells in the spleens of T-bet-cre$^{ERT2}$ x Rosa26$^{eYFP}$ mice following CGS-21680 administration, but no change in the frequency of eYFP$^+$ B cells in the lymph nodes, liver, and bone marrow or in the frequency of IgM ASCs in the bone marrow (Fig. 1c and Supplementary Fig. 2B). The data suggest that splenic T-bet$^+$ B cells did not migrate to other tissues or change their phenotype following CGS-21680 treatment. These data also indicate that splenic CD11c$^+$T-bet$^+$ B cells are the primary targets of the agonist, perhaps due to differences in A$_{2A}$ receptor expression between CD11c$^+$T-bet$^+$ B cells residing at different anatomical sites, differences in external signals received by the B cells, or to inherent differences in the ability of CD11c$^+$T-bet$^+$ B cells at various sites to respond to A$_{2A}$ receptor stimulation.

While the data indicate that CD11c$^+$T-bet$^{hi}$ B cells were depleted following CGS-21680 treatment, it was possible that the depletion was mediated indirectly. For example, previous studies have demonstrated that A$_{2A}$ receptor stimulation could inhibit the formation of GC B cells indirectly, by depleting T$_{FH}$ cells[39]. Therefore, we next utilized Mb1$^{cre/+}$ x Adora2a$^{flox/flox}$ mice, wherein the A$_{2A}$ receptor was deleted only in B cells. CD11c$^+$T-bet$^{hi}$ B cells were unaffected in Mb1$^{cre/+}$ x Adora2a$^{flox/flox}$ mice following CGS-21680 treatment, indicating that A$_{2A}$ receptor-mediated depletion of CD11c$^+$T-bet$^{hi}$ B cells required direct interaction of the agonist with the A$_{2A}$ receptor on B cells (Fig. 1d and Supplementary Fig. 3A). In contrast, CD11c$^+$T-bet$^{hi}$ B cells were depleted in CD4$^{cre/+}$ x Adora2a$^{flox/flox}$ mice that had been treated with CGS-21680 from days 30–37 post-infection, indicating that depletion of CD11c$^+$T-bet$^{hi}$ B cells did not require A$_{2A}$ receptor signaling on CD4$^+$ T cells (Fig. 1e and Supplementary Fig. 3B). Consistent with these observations, type 1 follicular helper cells (T$_{FH1}$) present on day 30 post-ehrlichial infection, possibly memory T$_{FH1}$ cells, were not depleted following CGS-21680

treatment (Supplementary Fig. 3C) These data indicate that A$_{2A}$ receptor agonists directly target CD11c$^+$T-bet$^+$ B cells.

**A$_{2A}$ receptor agonism depleted CD11c$^+$T-bet$^+$ B cells in lupus-prone mice.** Our studies of *E. muris*-infected mice suggested that CGS-21680 treatment may be useful for targeting CD11c$^+$T-bet$^+$ B cells in other contexts where these cells are pathogenic. Prior research in both mice and humans, has demonstrated that CD11c$^+$T-bet$^+$ B cells are generated during SLE and that these SLE-derived CD11c$^+$T-bet$^+$ B cells appear similar, if not identical, to CD11c$^+$T-bet$^+$ B cells that arise during *E. muris* infection[10,16,26]. T-bet-expressing B cells are thought to contribute to disease pathogenesis via the production of autoantibodies and/or via antigen presentation, and elimination of B cell-specific T-bet expression ameliorated disease in lupus-prone mice[26,51]. Therefore, we next addressed the effects of A$_{2A}$ receptor agonism on CD11c$^+$T-bet$^+$ B cells generated during autoimmunity using MRL/lpr (MRL/MpJ-Fas$^{lpr}$/J) mice, which provide a well-developed polygenic model for SLE[52]. CD11c$^+$T-bet$^+$ B cells were substantially reduced in the spleens of 20-week-old MRL/lpr mice that were treated with CGS-21680 twice weekly starting at 8 weeks of age (Fig. 2a). CGS-21680 treatment also significantly reduced the number of CD138$^+$B220$^+$ and CD138$^+$B220-negative cells (populations likely to include PBs, PCs, and perhaps double-negative T cells) in the spleens and lymph nodes of the same MRL/lpr mice (Fig. 2b). Given that T$_{FH}$ cells can contribute to the pathogenesis of SLE, as well as previous research indicating that autoimmune-derived T$_{FH}$ cells are susceptible to A$_{2A}$ receptor-mediated depletion, we analyzed spleens from MRL/lpr mice for the presence of CXCR5$^+$PD1$^+$CD4$^+$ T cells. We observed a decrease in CXCR5$^+$PD1$^+$CD4$^+$ T cells in CGS-21680-treated MRL/lpr mice (Fig. 2c). We did not observe an increase in the frequency of FoxP3$^+$ T$_{Reg}$ cells; the number of these cells was decreased in CGS-21680-treated mice (Supplementary Fig. 4A). Although CGS-21680 treatment did not alter the frequency of B cells or CD4$^+$ T cells, the number of these cells, as well as the total number of splenocytes, was reduced in CGS-21680-treated mice, indicating that CGS-21680 either directly or indirectly impacted a range of different lymphocytes in MRL/lpr mice (Supplementary Fig. 4B). Because no single experimental model fully recapitulates human SLE, we performed similar studies using SLE1.2.3. B6.NZM$^{Sle1/Sle2/Sle3}$ mice, which carry three SLE susceptibility alleles[53]. CGS-21680 treatment significantly reduced the number of T-bet$^+$ B cells in the spleens of 9-month-old SLE1.2.3 mice treated every other day for 5 days (Supplementary

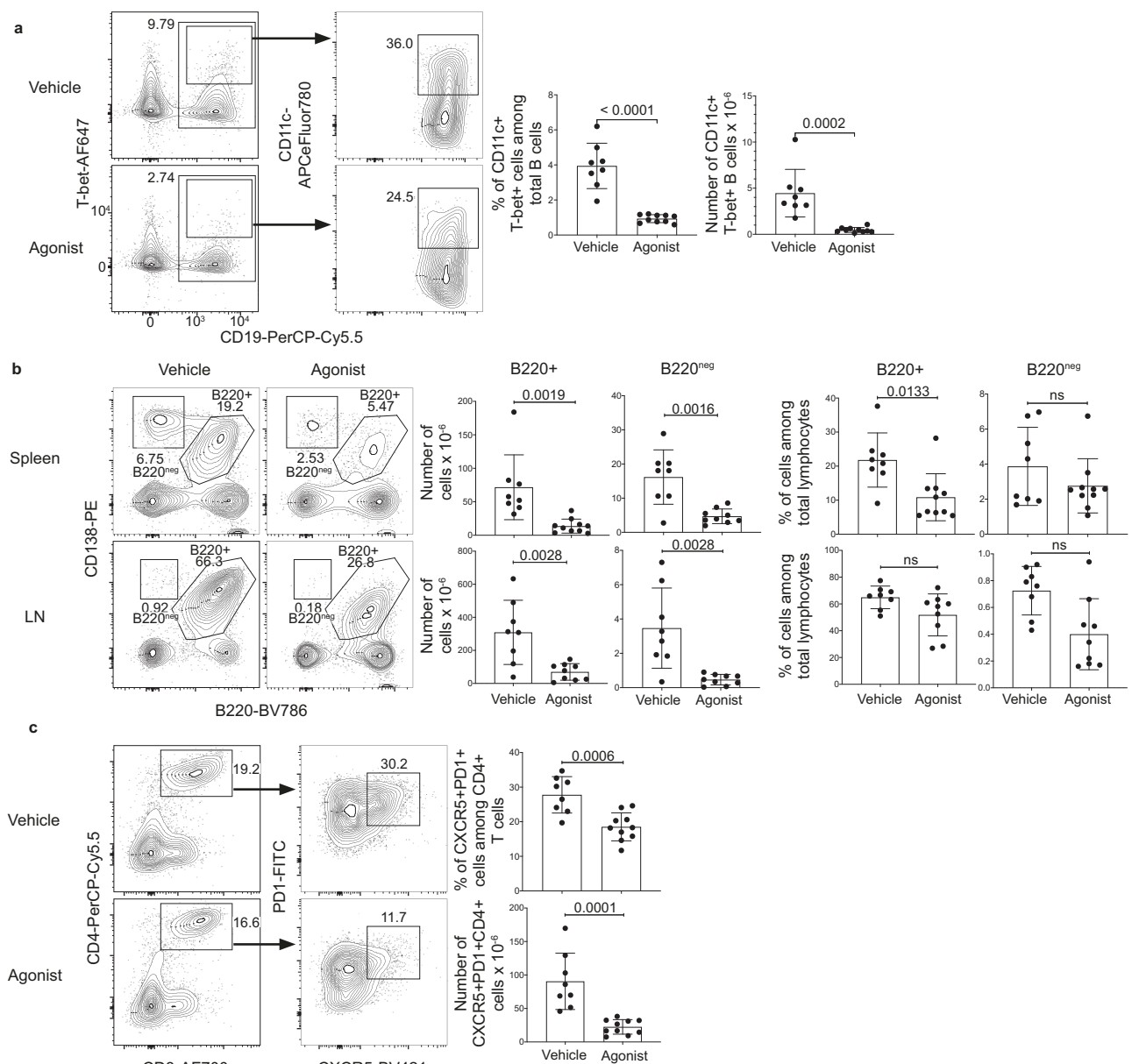

**Fig. 2 A$_{2A}$ receptor agonism depleted CD11c$^+$T-bet$^+$ B cells in lupus-prone mice. a** Female MRL/lpr mice were treated with vehicle ($n = 8$) or agonist ($n = 10$) twice weekly, starting at 8 weeks of age; splenocytes were analyzed at 20 weeks of age. The zebra plots and graphs show the percentages and numbers of T-bet$^+$CD11c$^+$CD19$^+$ B cells (%: $p < 0.0001$; #: $p = 0.0002$). Statistical significance was determined using either a two-tailed *unpaired t*-test (left graph) or a Mann–Whitney test (right graph). **b** Splenocytes and LNs from the mice in **a** were analyzed by flow cytometry. The contour plots and graphs show the percentages and numbers of CD138$^+$B220$^+$ cells and CD138$^+$B220-negative cells in the spleens and LNs of analyzed mice. Statistical significance was determined using two-tailed *unpaired t*-tests with the Benjamini, Kieger, and Yekutieli two-stage step-up false discovery rate method. The q values are shown in the graphs. **c** Splenocytes from the mice in **a** were analyzed by flow cytometry. The contour plots and graphs show the percentages and numbers of PD-1$^+$CXCR5$^+$CD4$^+$CD3$^+$ cells. Statistical significance was determined using two-tailed unpaired *t*-tests. Columns and error bars indicate the arithmetic mean and SD.

Fig. S4C). This short-term treatment did not result in the reduction of ASCs, as was observed following long-term treatment of MRL/lpr mice, although differences in treatment regimens complicate additional comparisons (Supplementary Fig. 4D). These data nevertheless indicate that A$_{2A}$ receptor agonists can deplete autoimmune-derived CD11c$^+$T-bet$^+$ B cells, although the treatment may target other lymphocytes as well.

**A$_{2A}$ receptor stimulation ameliorated disease in lupus-prone mice.** Previous studies have demonstrated that CD11c$^+$T-bet$^+$ B cells can contribute to SLE pathogenesis[26]. We therefore addressed whether A$_{2A}$ receptor-mediated depletion of CD11c$^+$ T-bet$^+$ B cells could ameliorate disease in a mouse model of lupus. We assessed disease severity following agonist treatment by analyzing the sera of 20-week-old MRL/lpr mice that had been treated with CGS-21680, twice weekly starting on week 8, for the presence of autoreactive antibodies. Anti-RNA and anti-dsDNA antibodies were reduced in MRL/lpr mice treated with CGS-21680, relative to vehicle-treated control mice, while anti-Smith antibodies were not significantly changed (Fig. 3a). IgG2a-specific anti-dsDNA antibodies were similarly reduced, although anti-RNA and anti-Smith

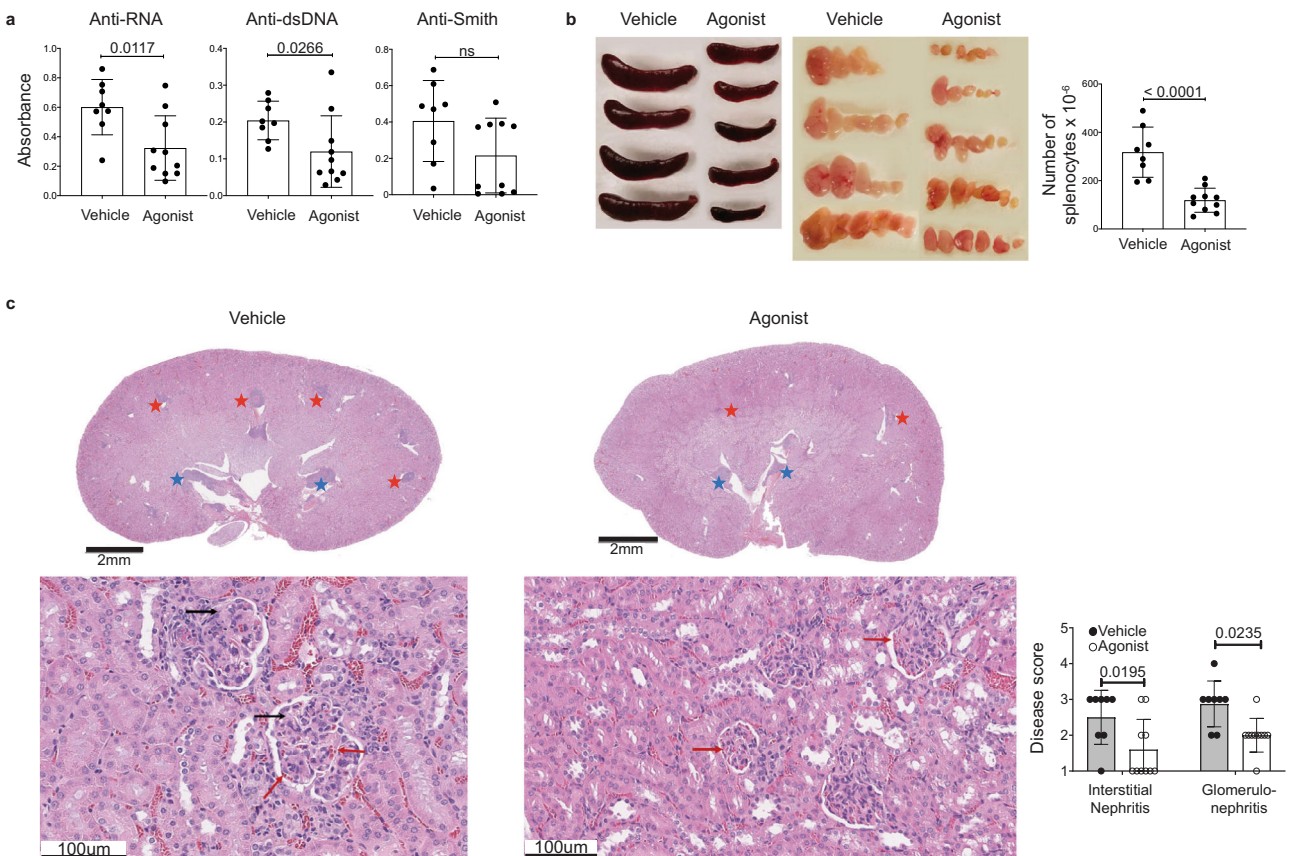

**Fig. 3 A$_{2A}$ receptor agonism ameliorated disease in a mouse model of lupus. a** Sera from the mice in Fig. 2a were analyzed for anti-RNA, anti-dsDNA, and anti-Smith antibodies by ELISA at 20 weeks of age. The graphs show relative absorbance values for the analyzed sera. Statistical significance was determined using a two-tailed unpaired t-test (left graph) or a Mann–Whitney test (middle and right graphs). **b** Spleens and LNs from the mice in Fig. 2a were analyzed at 20 weeks of age. The graph shows the number of splenocytes. Statistical significance was determined using a two-tailed unpaired t-test ($p < 0.0001$). **c** Kidneys from the mice in Fig. 2a were H&E-stained and were scored blindly for interstitial nephritis and glomerulonephritis at 20 weeks of age. Representative sections are shown (top: 1x, bottom: 20x). Lymphocyte and plasma cell aggregates are shown in the cortical medullary junction (red stars) and in the renal pelvis (blue stars). Patent (red arrows) and occluded capillary loops (black arrows) are shown. The graphs show the pathology scores. Statistical significance was determined using an ordinary two-way ANOVA with Sidak's multiple comparisons test. Columns and error bars indicate the arithmetic mean and SD.

IgG2a were unchanged (Supplementary Fig. 5A). The reduction of anti-dsDNA IgG2a is consistent with a decrease in T-bet$^{+}$ B cells, which characteristically undergo IgG2a switching[54]. Consistent with a reduction in CD138$^{+}$ cells and auto-antibodies, both lymphadenopathy and splenomegaly were reduced in CGS-21680-treated MRL/lpr mice compared to vehicle-treated controls (Fig. 3b). A$_{2A}$ receptor agonist treatment also reduced kidney pathology in MRL/lpr mice; both glomerulonephritis and interstitial nephritis were less severe in CGS-21680-treated MRL/lpr mice, compared to vehicle-treated controls (Fig. 3c). Kidneys from CGS-21680-treated MRL/lpr mice also contained fewer infiltrating lymphocytes. In contrast with previous studies, we did not observe a significant change in proteinuria in mice treated with CGS-21680 compared to controls ($p = 0.1694$; statistical significance was determined using a mixed-effect model with Sidak's multiple comparisons test) (Supplementary Fig. 5B). Future work will help resolve this disparity. CGS-21680 treatment modestly improved survival time among MRL/lpr mice, with a median survival time of 21.8 weeks among vehicle-treated mice, compared to a median survival time of 32.1 weeks among agonist-treated mice, although this difference was not statistically significant ($p = 0.535$). This modest effect was likely due to the inherent variability among MRL/lpr mice, the contribution of other A$_{2A}$ receptor tolerant cell types to disease, or to A$_{2A}$ receptor desensitization following

repeated agonist treatment (Supplementary Fig. 5C). Weight was not assessed as a measure of disease outcome, as CGS-21680 is known to affect weight gain in mice via an unrelated mechanism[55]. These data indicate that A$_{2A}$ receptor agonist treatment significantly improves disease outcome in a mouse model of lupus, likely in part due to the depletion of CD11c$^{+}$T-bet$^{+}$ B cells.

**A$_{2A}$ receptor agonism reduced the number of pathogenic lymphocytes after disease onset.** As prophylactic A$_{2A}$ receptor stimulation reduced the number of pathogenic lymphocytes and ameliorated lupus disease severity, we next assessed the therapeutic potential of A$_{2A}$ receptor stimulation on lupus after disease onset, a more relevant scenario for the treatment of SLE in humans. For these studies, MRL/lpr mice were treated with CGS-21680 twice weekly starting at 12 weeks of age, after disease onset, until 20 weeks of age, when the mice were sacrificed. A$_{2A}$ receptor stimulation significantly reduced the number of CD11c$^{+}$T-bet$^{+}$ B cells, although the percentage of these cells was not significantly changed (Fig. 4a). Moreover, splenic CD138$^{+}$ B220-negative and CD138$^{+}$ B220$^{+}$ B cells were reduced following CGS-21680 administration (Fig. 4b). In contrast to prophylactic A$_{2A}$ receptor treatment, CXCR5$^{+}$ PD1$^{+}$ T cells were not depleted following delayed CGS-21680 treatment (Fig. 4c). Moreover, the total

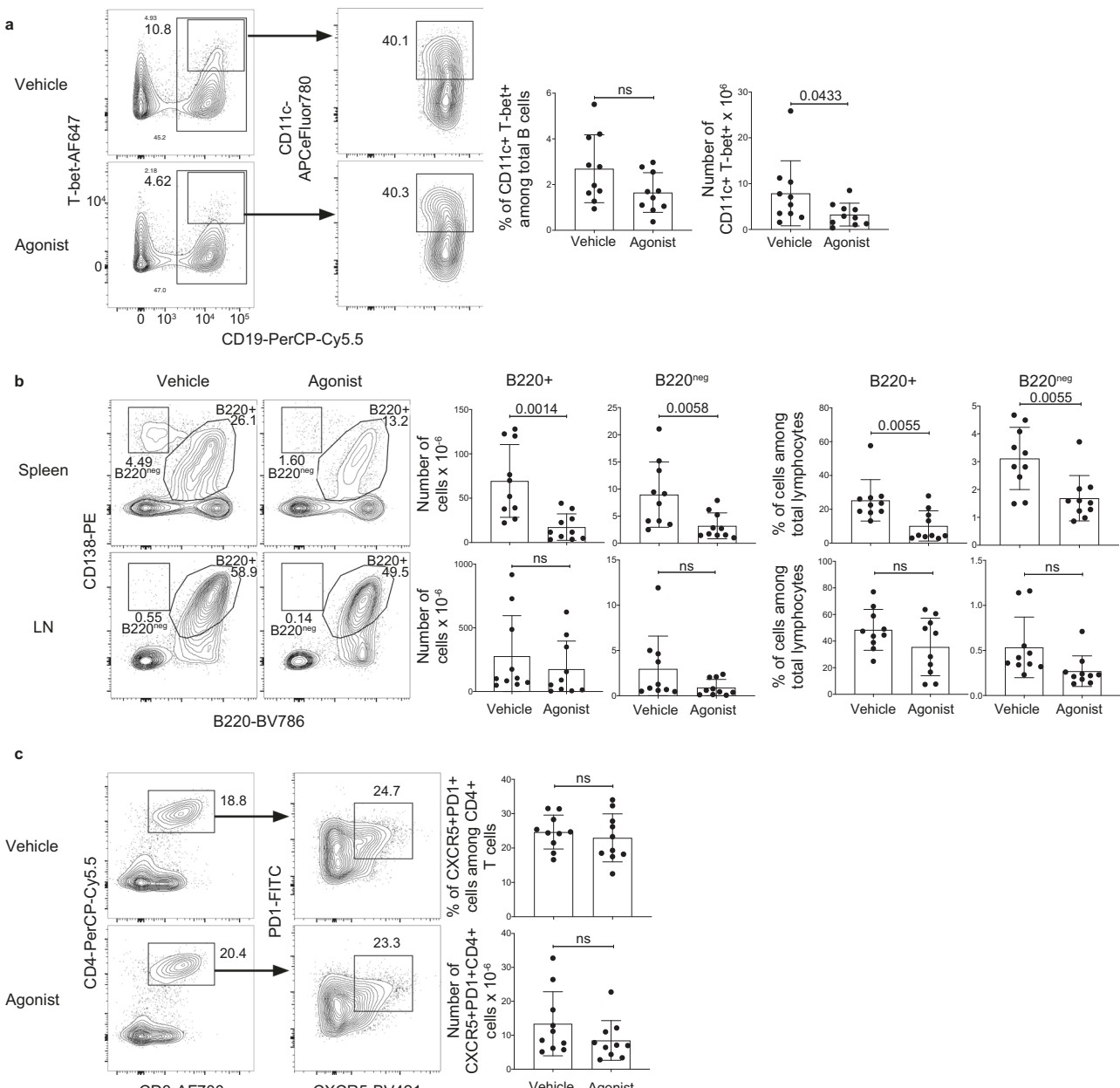

**Fig. 4 $A_{2A}$ receptor agonism reduced the number of pathogenic lymphocytes after disease onset. a** Female MRL/lpr mice were treated with vehicle ($n = 10$) or agonist ($n = 10$) twice weekly, starting at 12 weeks of age; splenocytes were analyzed at 20 weeks of age. The zebra plots and graphs show the percentages and numbers of T-bet$^+$CD11c$^+$CD19$^+$ B cells. Statistical significance was determined using either a two-tailed *unpaired t*-test (left graph) or a Mann–Whitney test (right graph). **b** Splenocytes and LNs from the mice in **a** were analyzed by flow cytometry. The contour plots and graphs show the percentages and numbers of CD138$^+$B220$^+$ cells and CD138$^+$B220-negative cells in the spleens and LNs of analyzed mice. Statistical significance was determined using two-tailed unpaired *t*-tests with the Benjamini, Kieger, and Yekutieli two-stage step-up false discovery rate method. *q*-values are shown in the graphs. **c** Splenocytes from the mice in **a** were analyzed by flow cytometry. The contour plots and graphs show the percentages and numbers of PD-1$^+$CXCR5$^+$CD4$^+$CD3$^+$ cells. Statistical significance was determined using a two-tailed *unpaired t*-test (top graph) or a Mann–Whitney test (bottom graph). Columns and error bars indicate the arithmetic mean and SD.

number of lymphocytes, B cells, and CD4$^+$ T cells were not affected by delayed CGS-21680 treatment (Supplementary Fig. 6A). These data indicate that $A_{2A}$ receptor stimulation can reduce the number of certain pathogenic lymphocytes even when administered after disease onset.

**$A_{2A}$ receptor stimulation after disease onset ameliorated disease in lupus-prone mice.** We next assessed the potential of $A_{2A}$ receptor stimulation as a treatment for SLE, by analyzing disease

severity in MRL/lpr mice that had been treated with CGS-21680 after disease onset. MRL/lpr mice were treated with CGS-21680 from weeks 12–20 of age and were sacrificed on week 20. This time point was chosen based on previously published studies[56,57]. Consistent with the reduced numbers of CD138$^+$ splenic cells, anti-dsDNA antibodies were significantly reduced in mice treated with CGS-21680 after disease onset, although anti-RNA and anti-Smith antibodies were unaffected (Fig. 5a). The reduction of anti-dsDNA antibodies and not anti-RNA antibodies could indicate that $A_{2A}$ receptor stimulation preferentially targeted plasmablasts

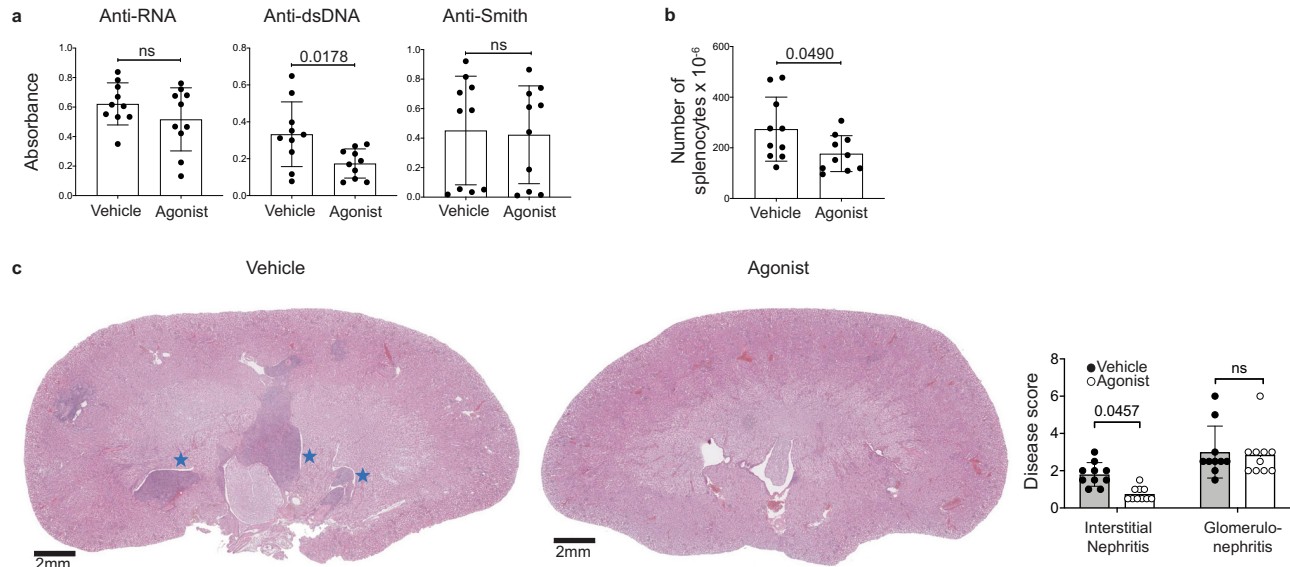

**Fig. 5 A$_{2A}$ receptor stimulation ameliorated disease in lupus-prone mice after disease onset. a** Sera from the mice in Fig. 4a were analyzed for anti-RNA, anti-dsDNA, and anti-Smith antibodies by ELISA. The graphs show relative absorbance values for the analyzed sera. Statistical significance was determined using a two-tailed unpaired $t$-test (left and right graphs) or a Mann–Whitney test (middle graph). **b** The graph shows the number of splenocytes from the mice in Fig. 4a. Statistical significance was determined using a two-tailed *unpaired* $t$-test. **c** Kidneys from the mice in Fig. 4a were H&E-stained and were scored blindly for interstitial nephritis and glomerulonephritis. Representative sections are shown (1x magnification). Lymphocyte and plasma cell aggregates are identified by blue stars. The graphs show the pathology scores. Statistical significance was determined using an ordinary two-way ANOVA with Sidak's multiple comparisons test. Columns and error bars indicate the arithmetic mean and SD.

and not long-lived plasma cells, as these subsets are thought to be enriched for anti-dsDNA and anti-RNA reactive B cells, respectively[56,57]. Similar to mice treated with CGS-21680 starting at 8 weeks of age, splenomegaly was reduced following delayed treatment as well (Fig. 5b). Delayed A$_{2A}$ receptor stimulation also significantly decreased interstitial nephritis, although glomerulonephritis remained unchanged (Fig. 5c), perhaps because the glomeruli were damaged prior to the start of treatment. These findings indicate that A$_{2A}$ receptor agonism can ameliorate SLE disease severity even after the onset of disease.

## Discussion

Although CD11c$^+$T-bet$^+$ B cells can function as memory cells and contribute to protective humoral immunity, this same B cell subset is also known to be involved in SLE and other autoimmune diseases[14–17,26]. Studies in mouse models of lupus have shown that elimination of T-bet expression in these B cells significantly reduced disease severity[26]. Consistent with these findings, we demonstrate that administration of the A$_{2A}$ receptor agonist CGS-21680 depleted CD11c$^+$T-bet$^+$ B cells generated in both *E. muris*-infected and lupus-prone mice, and reduced disease burden in autoimmune-prone mice. Our data are supported by a previous study that demonstrated that A$_{2A}$ receptor agonism reduced renal disease severity in a mouse model of lupus[49]. Thus, the targeted depletion of CD11c$^+$T-bet$^+$ B cells is likely a major mechanism by which A$_{2A}$ receptor stimulation ameliorates disease. We also show that A$_{2A}$ receptor stimulation can ameliorate disease in MRL/lpr mice after the onset of disease. Although A$_{2A}$ receptor stimulation after disease onset had less pronounced effects on CD11c$^+$T-bet$^+$ B cells, we nevertheless observed a decrease in the total number of these cells and a reduction in disease pathology. These findings indicate that A$_{2A}$ receptor stimulation can ameliorate SLE pathology, even after disease onset, supporting the use of A$_{2A}$ receptor agonists for the treatment of established SLE in humans. Our findings also suggest that A$_{2A}$ receptor agonist-mediated depletion of CD11c$^+$T-bet$^+$ B

cells will alleviate disease burden in other illnesses where CD11c$^+$ T-bet$^+$ B cells have been implicated in disease pathogenesis, such as RA, MS, and Sjögren's Syndrome[14,15,58–61]. Indeed, previous work has demonstrated that CGS-21680 treatment reduced the auto-antibody response and disease burden in a mouse model of RA[40]. In addition, the effects of A$_{2A}$ receptor signaling in CD11c$^+$T-bet$^+$ B cells could be important for other disease contexts such as chronic viral and bacterial infections in humans. Thus, pharmacological targeting of the A$_{2A}$ receptor on CD11c$^+$T-bet$^+$ B cells may provide a therapeutic approach for the treatment of SLE and other autoimmune diseases.

CGS-21680 was chosen for our studies as this drug is a well-documented and selective agonist for the A$_{2A}$ receptor, although it has been reported to interact with the A$_1$ receptor and A$_3$ receptor with lower affinity[62]. Using Mb1-cre x Adora2a$^{flox/flox}$ mice, we showed that CGS-21680 acts on T-bet$^+$ B cells via the A$_{2A}$ receptor. Additional studies will be required to corroborate these results in the MRL/lpr mouse model, although our data strongly suggest that these effects are mediated solely by the A$_{2A}$ receptor in those mice as well.

Our work demonstrates that CGS-21680-mediated depletion of CD11c$^+$T-bet$^+$ B cells requires expression of the A$_{2A}$ receptor on B cells generated during ehrlichial infection. This finding indicates that CGS-21680 acts directly on CD11c$^+$T-bet$^+$ B cells. Nevertheless, it is formally possible that A$_{2A}$ receptor agonists also acts indirectly, by targeting T$_{FH}$ cells, which have been shown to be critical for the generation of CD11c$^+$T-bet$^+$ B cells[5]. CXCR5$^+$PD1$^+$CD4$^+$ T cells were detected at much lower frequencies in lupus-prone MRL/lpr mice that had been treated from eight to 20 weeks of age, However, these T cells were not affected in MRL/lpr mice treated with CGS-21680 after disease onset. These data suggest that A$_{2A}$ receptor agonism impairs T$_{FH}$ cell differentiation, but does not deplete mature T$_{FH}$ cells[39,40]. Ongoing studies will resolve whether agonist administration in lupus-prone mice affects the function of fully mature T$_{FH}$ cells, and whether A$_{2A}$ receptor agonists directly target SLE-derived

$CD11c^+T-bet^+$ B cells. Regardless of the mechanism whereby $CD11c^+T-bet^+$ B cells are depleted following $A_{2A}$ receptor agonist administration, our work demonstrates that these cells are eliminated following treatment, and that this is accompanied by a reduction in disease burden.

In addition to $T_{FH}$ cells and $CD11c^+T-bet^+$ B cells, the $A_{2A}$ receptor is known to be widely expressed among other leukocyte populations[33,35]. It is, therefore, possible that CGS-21680 treatment ameliorates disease in MRL/lpr mice by acting on yet other cell types, in addition to $CD11c^+T-bet^+$ B cells. For example, $A_{2A}$ receptor stimulation may ameliorate disease also by inhibiting the differentiation of $T_{eff}$ cells, by abrogating the secretion of inflammatory cytokines by NK cells and T cells, or by directly acting on ASCs[49,63–65]. Similar to previous studies in a mouse model of RA, CGS-21680 treatment appeared to inhibit the generation of $T_{FH}$ cells in MRL/lpr mice, which could in turn impair the generation of autoreactive ASCs[40]. In our studies, CGS-21680 treatment significantly reduced the number of total splenic B cells and $CD4^+$ T cells in MRL/lpr mice, indicating either that $A_{2A}$ receptor agonism affects a wide range of cells, or that CGS-21680 treatment mitigated the inflammatory milieu driving lymphocyte proliferation. Thus, additional studies are required to formally address the cellular mechanisms whereby $A_{2A}$ receptor stimulation improves disease outcomes in lupus-prone mice. However, given that they contribute to the pathogenesis of SLE, and their clear susceptibility to $A_{2A}$ receptor-mediated elimination, we propose that depletion of $CD11c^+T-bet^+$ B cells is a major mechanism by which CGS-21680 treatment reduces disease in MRL/lpr mice.

We envision that the depletion of $CD11c^+T-bet^+$ B cells following CGS-21680 administration ameliorates disease in MRL/lpr mice via several possible mechanisms. First, we have previously demonstrated that $CD11c^+T-bet^+$ B cells can function as memory cells and can differentiate into ASCs, and other laboratories have shown that SLE-derived human $CD11c^+T-bet^+$ B cells can function as pre-ASCs[9,16,27]. Therefore, the loss of $CD11c^+T-bet^+$ B cells may indirectly reduce the number of autoreactive ASCs, thereby lowering the production of autoreactive antibodies. This reduction in autoreactive antibodies could in turn reduce immune complex deposition in the kidneys of MRL/lpr mice, and subsequently, reduce the number of kidney-infiltrating proinflammatory T cells. Second, as $CD11c^+T-bet^+$ B cells are known to function as potent APCs, the loss of these B cells following $A_{2A}$ receptor agonist treatment may also inhibit the activation of $CD4^+$ T cells, thus reducing T cell-mediated pathogenesis[28]. Indeed, while MRL/lpr mice unable to secrete antibodies still develop the disease, the loss of MHCII in B cells substantially improves disease outcome in MRL/lpr mice[51]. This suggests that antigen presentation, rather than antibody secretion, is a mechanism by which B cells contribute to SLE pathogenesis. Finally, depletion of $CD11c^+T-bet^+$ B cells may directly reduce inflammation in MRL/lpr mice, as SLE-derived $T-bet^+$ B cells have been shown to secrete proinflammatory cytokines[66].

While $A_{2A}$ receptor signaling has not been well studied in B cells, it has been characterized in other leukocytes, in particular in T cells. In $T_{eff}$ cells, $A_{2A}$ receptor stimulation results in the activation of PKA, which then inhibits ZAP70, various MAP kinases, and PKC, as well as Notch1 signaling, ultimately blocking T cell activation and the generation of IFN-γ and Granzyme B[67]. Other studies have also identified pAKT and NFκB as important components of $A_{2A}$ receptor signaling in T cells[34,65,68]. It is likely that the $A_{2A}$ receptor signals in B cells via similar intermediates, including pAKT, PKA, and NFκB, but the precise mechanism by which these signaling components induce cell death in B cells is unclear. $A_{2A}$ receptor agonism is known to downregulate MCL-1, an anti-apoptotic protein required for the maintenance of activated B cells and PCs[69–72]. MCL-1 binds to BH3-only proteins

and prevents them from activating Bax or Bad, and inducing apoptosis[73]. It is possible that $A_{2A}$ receptor agonism induces cell death in $CD11c^+T-bet^+$ B cells by inhibiting MCL-1 and allowing BH3-only proteins to activate Bax and Bak, ultimately allowing for the release of cytochrome c and the induction of apoptosis[69,70,73].

In our preliminary studies, we failed to observe any changes in the phenotype or function of $T-bet^+$ B cells that lacked expression of the $A_{2A}$ receptor. Neither administration of the $A_{2A}$ receptor antagonist Istradefylline, nor genetic elimination of $A_{2A}$ receptor, affected the development of $CD11c^+T-bet^+$ B cells in E. muris-infected mice. Although well studied in other cells, $A_{2A}$ receptor signaling in normal B cells is not well understood. Studies in CD73-deficient mice demonstrated that loss of this ectoenzyme did not inhibit the formation of GC or memory B cells, but did impede the maintenance of bone marrow PCs[74]. Given the well-documented immunosuppressive function of $A_{2A}$ receptor signaling in other cell types, $A_{2A}$ receptor signaling is likely similarly suppressive in B cells. The limited effects of $A_{2A}$ receptor depletion in our studies suggest that there may be minimal $A_{2A}$ receptor signaling under physiological conditions. Therefore, $A_{2A}$ receptor signaling may only occur during specific instances characterized by high levels of extracellular adenosine, such as during hypoxia or inflammation. It is also possible that $A_{2A}$ receptor signaling in $T-bet^+$ B cells is a redundant regulatory mechanism and is only necessary in the absence of other regulatory signals.

Strategies to target B cells have been effective at ameliorating disease in RA and MS, although the efficacy of such methods in the treatment of SLE and yet other autoimmune diseases remains uncertain[75,76]. Two clinical trials failed to show a significant reduction in symptoms in SLE patients treated with the B cell-depleting antibody, Rituximab (i.e., anti-CD20). However, these findings may have been confounded by heterogeneity within the sample groups, incomplete B cell depletion, and by the high levels of corticosteroids administered to both groups of patients[77]. Current B cell-targeted treatments lack specificity, and act by eliminating most B cells in patients (i.e., 90–100% of peripheral B cells)[78]. Pan-depletion of B cells also eliminates suppressive B cells, and it has been proposed that loss of these B cells contributes to the spike in symptoms observed among some MS patients receiving anti-B cell therapy[79]. Because of the adverse effects of current B cell-targeted therapies, there is a need for novel treatments with limited and tolerable side-effects. Therapies that target the adenosine signaling pathway may therefore provide a promising avenue for treatment of SLE and other autoimmune diseases. Indeed, the $A_{2A}$ receptor agonist Regadenoson (also known as Lexiscan) is already used for myocardial imaging in patients unable to undergo exercise-induced stress testing[80]. Because of its wide use, side effects from Regadenoson treatment have been thoroughly documented and are well-tolerated in humans, at least under the conditions of current usage. Side effects include, headache, chest pain, dizziness, and disruption of the blood-brain barrier, although most of these symptoms resolve within 15 min following treatment[81]. Therapeutic treatment for autoimmunity will likely require repeated agonist administrations, and although the dose and frequency of administration required to limit disease in humans is not yet known, modest side effects may be well-tolerated in otherwise healthy patients. Additional research is required to determine the efficacy of $A_{2A}$ receptor agonists in humans, but our research suggests that this class of drugs represents a promising approach for the treatment of autoimmune diseases caused in part by $CD11c^+T-bet^+$ B cells.

## Methods

**Mice.** C57BL/6J, CD4$^{cre}$ (B6.Cg-Tg(Cd4-cre)1Cwi/BfluJ), Mb1$^{cre}$ (B6.C(Cg)-Cd79a$^{tm1(cre)Reth}$/EhobJ), MRL/lpr (MRL/MpJ-Fas$^{lpr}$/J), SLE123 (B6;NZM-

$Sle1^{NZM2410/Aeg}$ $Sle2^{NZM2410/Aeg}$ $Sle3^{NZM2410/Aeg}$/LmoJ), and Rosa26$^{eYFP}$ (B6.Cg-Gt(ROSA)26Sor$^{tm3(CAG-EYFP)Hze}$/J) mice were obtained from The Jackson Laboratory (Bar Harbor, ME). T-bet-cre$^{ERT2}$ mice were generated by Dr. Lin Gan at the University of Rochester, Rochester, NY. Adora2a$^{flox}$ (B6;129-Adora2a$^{tm1Dyj}$/J) mice were provided by Dr. Joel Linden, La Jolla Institute for Immunology, La Jolla, CA. All mice, except for the MRL/MpJ-Fas$^{lpr}$/J, were on the C57BL/6J background. All mice were housed and bred in the SUNY Upstate Medical University Animal Care Facility (Syracuse, NY), in accordance with institutional guidelines for animal welfare. All mice used for experiments were at least 6 weeks old, and both male and female mice were used unless otherwise stated. All studies involving animals were approved by the SUNY Upstate Medical University Institutional Animal Care and Use Committee NYSDOH Unit A073 IACUC number 311.

**Infections and drug administration.** Mice were infected intraperitoneally with $5–10 \times 10^4$ E. muris bacterial copies, as determined by qPCR, and as previously described (probe sequence: 56-FAM/AGGGATTTC/ZEN/CCTATACTCGGT/3IABkFQ)[82]. CGS-21680 hydrochloride and Istradefylline were purchased from Cayman Chemical (Ann Arbor, MI). Regadenoson (Lexiscan; Astellas Pharma Inc.) was purchased internally from SUNY Upstate Medical University. E. muris-infected and lupus-prone mice were injected intraperitoneally with 50 μg of CGS-21680 (C57BL6/J: 2.5 mg/kg; MRL/lpr: 1.43 mg/kg), 132 μg of Istradefylline (6.6 mg/kg), or 50 μg of Regadenoson (2.5 mg/kg) in 200 μl PBS containing 5% DMSO, without anesthesia.

**Flow cytometry and antibodies.** Spleens and lymph nodes were disaggregated using a 70 μm cell strainer (BD Falcon). Erythrocytes were removed by incubation with ACK lysis Buffer (Quality Biological Inc). Cells were treated with anti-CD16/32 (2.4G2) and mouse cells were incubated with the following antibodies: PerCpCy5.5-conjugated anti-CD19 (6D5, 1:200), Alexa Fluor 700-conjugated anti-CD19 (6D5, 1:200), APC-eFluor 780-conjugated anti-CD11c (N418, 1:200), Brilliant Violet 785-conjugated anti-B220 (RA3-6B2, 1:200), V500-conjugated anti-B220 (RA3-6B2, 1:100), Alexa Fluor 647-conjugated anti-T-bet (4B10, 1:400), PerCpCy5.5-conjugated anti-T-bet (4B10, 1:200), FITC-conjugated anti-PD-1 (29F.1A12, 1:100), PerCPCy5.5-conjugated anti-CD4 (RM4-4, 1:200), Alexa Fluor 700-conjugated anti-CD3 (17A2, 1:400), Brilliant Violet 421-conjugated anti-CXCR5 (L138D7, 1:200), PE-conjugated anti-CD138 (281–2, 1:200), Brilliant Violet 421-conjugated anti-IgM (RMM-1, 1:200), PE-conjugated anti-FoxP3 (150D, 1:200). LIVE/DEAD Fixable Aqua Dead Cell Stain (Thermo Fisher Scientific) or Ghost Dye Violet 510 (TONBO biosciences) were used to stain for viability.

The cells were stained at 4 °C for 30 min, washed, and analyzed. For intracellular staining, surface-stained cells were fixed/permeabilized for 40 min at 4 °C, using the Transcription Factor Buffer set Fixation/permeabilization buffer (BD Pharmingen), washed, stained at 4 °C for 30 min, washed, and analyzed. Unstained cells were used to establish the flow cytometer voltage settings, and single-color positive controls were used to adjust compensation. The data were acquired on a BD Fortessa flow cytometer using Diva software (BD Bioscience) and were analyzed with FlowJo software version 10.7 (BD Bioscience.). All flow cytometry analyses were performed on freshly harvested cells.

Livers were perfused with PBS and disaggregated using a 70 μm cell strainer (BD Falcon). Cells were gradient separated using 40% Percoll (Sigma Aldrich) and erythrocytes were removed by incubation with ACK lysis Buffer (Quality Biological Inc).

**Proteinuria.** Urine albumin concentration was determined using Fisherbrand 10-SG Urine Reagent Strips according to the manufacturer's instructions.

**Histology.** Kidneys were harvested from mice and fixed in 4% PFA for at least 48 h at room temperature. Fixed kidneys were paraffin embedded, cut into 5 μm sections, stained with hematoxylin and eosin (H&E), and blindly scored by a single pathologist at Histowiz (Brooklyn, NY) or by a certified pathologist at the University of Pittsburgh Medical Center, using a previously described metric[83]. Gomerulonephritis was assessed using a scale from one to six based on mesangial cellularity and expansion, the presence of patent capillary loops, glomeruli size, karyorrhexis, crescent formation, and sclerosis. Interstitial nephritis was graded on a scale from one to four based on the prevalence of lymphocytes and plasma cell infiltrates in the perivascular area and/or in the interstitial space.

**ELISAs.** Anti-RNA ELISAs were performed as described in Blanco et al.[84]. Flat-Bottom Immuno plates (Thermo Scientific) were coated with 100 μl of poly-L-lysine (50 μg/ml) (Sigma Aldrich), followed by 100 μl of yeast RNA (15 μl/ml) (Fisher Scientific). The plates were blocked with fetal bovine serum and incubated with serum (1:100 dilution in wash buffer), overnight at 4 °C. RNA-specific antibodies were detected with alkaline phosphatase-conjugated goat anti-mouse antibodies (1:1000) and background removed using no serum controls. (Southern Biotechnology Associates, Birmingham, AL).

Anti-Smith and Anti-dsDNA ELISAs were performed using the Bio-Rad Anti-Sm (cat # 96SM) and Anti-dsDNA (cat # 96DS) EIA Kits (Bio-Rad), according to the manufacturer's instructions, except that anti-Smith and anti-dsDNA antibodies

were detected using alkaline phosphatase-conjugated goat anti-mouse secondary antibodies (1:1000) (Southern Biotechnology Associates, Birmingham, AL).

**Statistical analysis.** Statistical analyses were performed using Prism 9 (Graph-Pad). The statistical tests that were performed are indicated in the figure legends. The column in each of the plots indicates the arithmetic mean of the dataset, and upper and lower bounds indicate the standard deviation of the dataset. Data were analyzed for normality using the Shapiro–Wilk test and statistical tests were chosen based on the normality of the data. An effect size of 2.02 was estimated based on previous data and assuming a power of 80% and a significance level of 0.05.

**Reporting summary.** Further information on research design is available in the Nature Research Reporting Summary linked to this article.

## Data availability

All other data are provided in the article and its Supplementary Files or from the corresponding author upon reasonable request. Source data are provided with this paper.

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

## Acknowledgements

We gratefully acknowledge the excellent technical assistance provided by the Upstate Medical University Flow Cytometry Core. We thank Lisa Phelps, and Joanne Chilton for their assistance, and Dongliang Wang for his advice on statistical analyses. We also thank Drs. E. Leadbetter (University of Texas Health Science Center) and J. Wilmore (SUNY Upstate Medical University) for their helpful reviews of the manuscript. This work was supported by the U.S. Department of Health and Human Services grant R01AI114545 awarded to G.M.W and AI072648 to A.P.

## Author contributions

Conceptualization, R.C.L. and G.M.W.; methodology, R.C.L. and G.M.W.; formal analysis, R.C.L.; investigation, R.C.L., K.L.N., B.C.-M., J.C., A.P., S.I.B., and G.M.W.; writing—original draft, R.C.L. and G.M.W.; writing—review and editing, R.C.L., G.M.W., and K.L.N.; visualization, R.C.L. and G.M.W.; supervision, G.M.W.; project administration, G.M.W.; funding acquisition, G.M.W.

## Competing interests
The authors declare the following competing interests: G.M.W. and R.C.L. are inventors listed on a patent pending for the use of $A_{2A}$ receptor agonists in the therapeutic depletion of CD11c[+]T-bet[+] B cells in diseases mediated by these B cells. Applicant: SUNY Research Foundation, Application Serial Number: PCT/US2019/045624. The remaining authors declare no competing interests.
