## [Peer Review File · Nature Communications]

Adenosine Receptor 2a Agonists Target Murine CD11c+T-bet+ B cells in Infection and AutoimmunityREVIEWER COMMENTS

Reviewer #1 (Remarks to the Author):

This study by Levack et al. focuses on an important therapeutic area, namely the pharmacological targeting of "age/autoimmunity associated B cells (ABCs)" in systemic autoimmunity. As the authors note, multiple studies over the past decade have reported that CD11c+Tbet+ B cells are increased in subjects with autoimmunity, that these cells likely contribute to disease manifestations, and that have defined the immune signals required for ABC formation. The overall model emerging from these studies (including prior work from the Winslow group) is that ABCs are a subset of memory B cells that can rapidly differentiate into plasma cells following secondary challenge. However, there are currently no effective strategies to deplete ABCs. Thus, it has been difficult to determine the relative contribution of ABCs (as opposed to other immune subsets) to the clinical manifestations of SLE. For this reason, the current study holds both mechanistic and potential translational significance.

First, the authors leverage published RNA-Seq data showing that A2aR expression is increased in *E. muris* infected CD11c+Tbet+ B cells. Next, they show that separate A2aR agonists deplete CD11c+Tbet+ B cells, via a mechanism likely related to induction of cell death, not cell migration or downregulation of CD11c/Tbet expression. Using Adora2a floxed mice, they confirm that A2aR agonists deplete CD11c+Tbet+ B cells in a B cell-intrinsic manner. Next, the authors confirm that A2aR agonists are able to deplete CD11c+Tbet+ B cells in separate murine lupus models, resulting in ameliorated disease. Finally, the investigators show that T-bet-expressing B cells in both healthy controls and lupus patients express A2aR, suggesting that A2aR agonism may be an effective therapy in human SLE.

Overall, these experiments are logically designed, and the data appropriately interpreted, with relevant caveats acknowledged in the discussion. While these observations are important, it should be noted that the findings involving CGS-21680 treatment of MRL/lpr mice are largely consistent with an earlier published study (Zhang, et al. *Lupus*, 2011). Thus, the major new finding of this current manuscript is that A2aR agonism ablates CD11c+Tbet+ B cells. The authors propose that this mechanism likely accounts for the therapeutic benefits of A2aR agonism in SLE, although definitive proof is lacking because of known impacts of A2aR agonists on other immune lineages (Tfh, etc.). For this reason, greater mechanistic understanding of how A2aR agonism promotes CD11c+Tbet+ B cell depletion in contrast with impacts on other immune cells implicated in lupus pathogenesis would strengthen this study.

Specific comments:

1. A2aR expression on murine B cell subsets: Are anti-murine A2aR antibodies available to measure surface A2aR expression on different B cell subsets following *Ehrlichia* infection and/or murine lupus models? The authors note that A2aR transcript is elevated on CD11c+Tbet+ B cells following *E. muris* infection, but don't confirm increased protein expression by flow cytometry in the murine studies (expression was only reported on human B cells). Given the distinct sensitivity of different B and T cell populations to A2aR agonists, it would be useful to compare A2aR expression on, for example, naïve B cells, CD11c+Tbet+ B cells, plasma cells, Tfh cells, Tregs, etc.
2. Although splenic CD11c+ B cells appear exquisitely sensitive to A2aR agonists, it is surprising that T-bet+ B cells in lymph nodes, liver, BM are resistant to CGS-21680 (Fig. 1C). Do the authors have a better explanation for this? What is A2aR surface expression on eYFP+ B cells at these different sites. Since T-bet is expressed by multiple B cell lineages (not just ABCs), is the phenotype of eYFP+ B cells different based on tissue location. Are all eYFP+ cells phenotypically ABCs (CD11b+CD11c+CD21 low, etc.)?
3. MRL.lpr model:
 - Autoantibodies: Since A2aR agonists predominantly deplete T-bet+ B cells, one would predict that IgG2a/c subclass autoantibodies would be most impacted by A2aR agonist treatment.

Additional analysis of autoantibody subclasses would thus support the investigators model in which CGS-21680 treatment predominantly targets CD11c+Tbet+ ABCs.

- Proteinuria: Despite statistically significant histologic decrease in glomerulonephritis, no difference was observed in degree of proteinuria. These appear predominantly driven by an absence of worsening proteinuria in the vehicle-treated animals, and contrasts with the findings of Zhang, et al. (Lupus, 2011). Do the authors have an explanation for this, since proteinuria is typically observed in the MRL.lpr model at this age. Were any other markers of renal dysfunction assessed (BUN, creatinine)?

- Mortality curve: Given the striking reduction in splenomegaly/lymphoid hyperplasia, and decrease in activated B and T cells following CGS-21680 treatment, it is surprising that all the treated animals succumbed from autoimmunity with a modest (non-significant) extension in median survival. Do the authors have an explanation for this? Could this be related to adenosine receptors desensitization following repeated agonist stimulation (which may limit human translational potential).

4. Human B cell A2aR expression:

- How does A2aR surface expression on Tbet-hi B cells compare with non-B cell lineages? For example, is A2aR expressed at higher levels on Tbet-hi B cells relative to activated memory T cells or CXCR5+ circulating Tfh cells? This would help to validate the idea that A2aR agonism primarily functions in SLE via the depletion of CD11c+Tbet+ memory B cells.

Reviewer #2 (Remarks to the Author):

This original research manuscript by Levack, Winslow and colleagues reports for the first time, to the best of my knowledge, that use of adenosine 2a receptor (A2aR) agonists depletes CD11c+ Tbet+ B cells in ehrlichial-infected and lupus-prone mice. CD11c+ Tbet+ B cells have emerging roles in maintaining inflammation during infection and autoimmunity, however the role of A2aR activation in these B cells is undefined (and even for other B cell subsets remains largely undefined). Importantly, this manuscript demonstrates that A2aR activation reduces histological and biochemical features of lupus in mice, although not clinical disease per se. Finally, this report confirms previous reports in mice, that human CD11c+ Tbet+ B cells express relatively high A2aR compared to other B cells. Collectively, this study represents a highly novel and potentially important contribution to these fields and the broader immunology research community, but a number of issues need to be addressed before acceptance can be considered.

Major comments

1. The Abstract claims to report the impact of A2aR activation on CD11c+ Tbet+ B cells however review of the data shows that either CD11c+ OR Tbet+ B cells (not CD11c+ Tbet+ B cells) were studied, presumably because they can be largely used interchangeably. The authors should comment on this in the manuscript.
2. Flow cytometry: The manuscript does not provide the full gating strategies used to study key murine and human immune cell populations (and as prescribed in the "Reporting Summary").
3. Statistical analysis: There is no comment of normality of data, which should be provided (and as prescribed in the "Reporting Summary").
4. Although this study uses the selective A2aR agonist Regadenosen and A2aR KO mice to confirm some findings, the study largely relies on the use of the A2aR agonist CGS-21680, which can also bind A1R and A3R albeit to a lesser extent (doi: 10.1007/s11302-015-9460-9). This reference and caveat should be noted in the manuscript.
5. Based on comparisons of lymphoid tissues and liver, lines 230-231 claim "that splenic CD11c+ Tbet+ B cells are the primary targets of the [CGS-21680]" however the authors provide no explanation as to why. The authors should comment on this in the manuscript.
6. Although the Discussion outlines future studies, has the impact of CGS-21680 on murine and human CD11c+ Tbet+ B cells in mediating cell death in vitro been studied? Given the in vivo

effects, even with a single dose (Fig 1B), it would appear relatively straight forward to incubate murine splenocytes and human PBMCs in vitro in the absence and presence of CGS-21680 and measure cell death in these and other cell subsets, and include this data in the manuscript.

Minor comments

1. The manuscript should adopt IUPHAR/BPS nomenclature for adenosine receptors (<https://www.guidetopharmacology.org/>).
2. Line 45: The capitalization of "Influenza" and "Hepatitis" appears unnecessary.
3. Line 49: Correct on first use "Ehrlichia muris" to "E. muris".
4. Lines 132-137: In addition to amount of drug per mouse, the Methods should also provide the approximate doses (approximate amounts/kg) for transparency and comparison to other studies. Further this text should clarify that these does refer to both the infection and lupus models.
5. CGS-21680 can reduce weight in other mouse models of disease to confound results (doi: 10.1016/j.intimp.2019.04.037 and 10.1007/s11064-019-02745-0). The authors should comment in the manuscript whether the CGS-21680 regime used in the present study also impacted weight.
6. Lines 139-161 and 163-169: The flow cytometry methodology is a little muddled by listing the anti-human antibodies amongst the anti-murine antibodies even though lines 139-161 only deals with murine spleen and lymph node cells to this point. These sections should be revised to improve clarity, perhaps by listing anti-human antibodies including the antiA2AR antibody (which seems to be the same for both mouse and human studies) at lines 163-169.
7. Line 174: correct "4%PFA" to "4% PFA".
8. Lines 296 and 299: Although in Fig 3E legend, it would be helpful if the P vales for these non-significant differences (Fig 3D and 3E) were stated in the text also.
9. Figure 3 panel: Time points of tissues analyses should be stated for 3A, 3B and 3C. Also add the P value in panel for Fig 3D as for 3E.
10. Supplementary data should be proof-read/grammar checked for occasional typographical errors (e.g. "Bone Marrow" instead of "Bone marrow" in Fig S1D).

Reviewer #3 (Remarks to the Author):

Summary

Overall Aim

This study aims to identify a selective and specific method of targeting Tbet+ CD11c+ B cells, a potentially pathogenic B cell population in SLE (and other autoimmune conditions). In doing so this could provide a new pharmacological target for the treatment of SLE that results in specific depletion of B cells that drive pathology, without depletion of the entire B cell compartment.

Earlier findings

Prior work from this group identified an enhanced expression of adenosine receptor 2a (A2aR) in Tbet+ CD11c+ B cells, distinguishing this subset from other B cell populations in mice.

Aim

In light of their previous finding, the authors sought to selectively deplete established Tbet+ CD11c+ B cells using A2aR agonists. Two different immunological mouse models known to involve Tbet+ B cell responses were chosen for A2aR agonist mediated ablation; Ehrlichia muris intracellular bacterial infection, and systemic lupus erythematosus (lupus).

Current Findings

Daily treatment of E.muris infected mice with A2aR agonist CGS-21680 resulted in significant depletion (75-80%) of CD19+ CD11c+ B cells and CD19+ Tbet+ B cells in the spleen. One dose of agonist was sufficient to deplete (80%) of the CD11c+ B cells (24hrs post administration). A conditional A2aR KOs using mb-1 cre abolished depletion after treatment with the agonist whereas this was not the case with a CD4 cre, supporting that this was the result of direct action of the A2aR agonist on B cells and not via Th cells.

Similar depletion of Tbet⁺ and CD11c⁺ B cells was found in Lupus prone mice, MRL/lpr, in response to long term A2aR agonist. However, this treatment regime also resulted in depleted plasma cells (PCs) and plasmablasts (PBs) in the spleen and LNs. In this instance Tfh cells were also depleted which likely contributes to limited PC/PB. The A2aR agonist treatment in lupus prone mice broadly targeted multiple lymphocytes. Long term treated mice experienced lower autoantibody titres, RNA and dsDNA but not Smith antigen, reduced splenomegaly and lymphadenopathy, and improved kidney pathology with fewer infiltrating lymphocytes whereas proteinuria and overall-survival were not altered.

Finally, the authors investigated whether Tbet⁺ B cells in PB from healthy donors (HD) and patients with SLE express A2aR. The authors conclude that they do and that the levels are lower in the latter.

General comments

- The study adheres to ethical standards including ethics committee approval and consent procedures. Ethical approval and written consent for human PB studies listed and animal welfare standards adhered to.
- Standard biosecurity and institutional safety procedures have been adhered to

Strengths

- The aims of this study are robust, clear, and have clinical relevance
- Well written and clear
- Good summary of the literature
- Clear reasoning for the need and therapeutic potential specific Tbet⁺ B cell targeted treatment.
- Valuable application of a target already pharmacologically approved.
- E.muris study strong in determining Tbet CD11c B cell depletion as direct action of the agonist on the B cells. Well explored possibility of alternate migration and impact on other lymphocytes.

Limitations

- Only N=5 for most murine studies.
- Mouse models only run once
- Only N=6 for the human studies. The HDs were 25 to 55 years whereas the SLE patients were 41-65 years. With this low number of samples in each group and with no correction for e.g., age, disease activity, etc it is not possible to draw any conclusions from the results presented

Bacterial-infection model

1. The elevated levels of A2aR expression in the bacterial-infection model relied on previous mRNA data. However, there are no data on protein levels, not on the CD11c⁺ / Tbet⁺ or other B cells, or other cell types in the mouse models. This is important as mRNA and protein do not necessarily have to correlate. Moreover, evidence of A2aR expression exclusively on Tbet⁺ B cells in the two models is lacking.
2. The development of Tbet⁺ CD11c⁺ B cells in the bacterial infection model is TLR7-independent whereas in the lupus model is TLR7-dependent. Hence, the Tbet⁺ CD11c⁺ B cells in the two models are not necessarily 'the same'. Are there any human chronic bacterial infection diseases where Tbet⁺ CD11c⁺ B cells accumulate? This would be more relevant than SLE and potentially very interesting.
3. Treatment of E.muris infected mice with A2aR agonist CGS-21680 apparently depletes CD19⁺ CD11c⁺ B cells and CD19⁺ Tbet⁺ B cells in the spleen. However, whilst still being referred to as Tbet⁺ CD11c⁺ B cells, the authors do not co-stain Tbet and CD11c⁺ on the B cells which is important to clearly demonstrate that it is the same population
4. Cell numbers of other B-cell subsets are not shown, hence potential effects on these are not clear
5. What impact does depletion of this cell type have on the infection clearance etc?

Lupus models

6. Depletion of Tbet⁺ and CD11c⁺ B cells was also found in Lupus prone MRL/lpr mice. However, it is unclear whether the observed changes can be attributed to the A2aR agonist treatment depleting of Tbet⁺ B cells, as other cell types were also affected by the treatment, and it appears

to be a more general immunosuppression in this disease setting (impact on plasma cells, Plasma blasts, and the T cells compartment).

7. It is no longer a targeted depletion. Therefore, interpretation of the follow up clinical pathology can no longer be attributed to Tbet+ B cells but could be the result of more general immunosuppression.

8. Anti-RNA and -DNA antibodies were reduced, however, these cannot be referred to as pathogenic, this was not tested and it is not known whether all e.g., anti-DNA antibodies are pathogenic

9. Whether there is a general decrease in antibody levels and if 'non-pathogenic' antibody production is also impacted was not tested

10. Authors conclude that proteinuria levels have a trending reduction in agonist treated mice. However, these groups are always overlapping, and hence there is no reduction, and it is stretching it to refer to as a trend

11. The overall-survival after treatment is not statistically significant, there are far too few animals to observe a potentially significant effect, if there is any at all

12. Whether A2aR expression is limited to Tbet+ B cells in the Lupus prone mice was not analysed. The referenced article which details A2aR enhanced expression is from E.muris infection. It would be necessary to explore A2aR expression on Tbet+ and Tbet- B cells in the lupus models, and on the T cells in these animals.

13. Another lupus model was also employed but with a different treatment regime. This makes it difficult to compare if the targeted effect of the A2aR agonist found in the short-term treatment was the result of the model itself or the reduced treatment time?

Human HD and SLE patients

14. In the figure (Fig. 4), 2.22% of peripheral blood B cells express Tbet. This population is not very distinct and the variation between donors is not shown. Maybe it varies a lot, which would make it difficult to gate these cells and it would affect gating of the cells and detection of A2aR considerably.

15. The gated Tbet+ cells from HD demonstrate a bi-modal pattern of A2aR expression in Tbet+ cells and a lack of expression in Tbet- cells. However, in the absence of an FMO, it is unclear whether there is a lack of expression or just lower levels in the Tbet- cells.

16. As the proportion of Tbet+ cells is low, and since the proportion in the other samples is unknown, the variation could stem from very few Tbet+ cells, and affect the variation substantially.

17. Adding to this the bi-modal A2aR expression pattern, which might be reflected in the variation between samples in terms of A2aR MFIs, 3-fold on Tbet- and 6-fold on Tbet+ cells. Thus, despite a calculated statistically significant difference, the data are not convincing.

18. Performing the same experiment on PB from patients with SLE showed a very low proportion (<1%) of Tbet+ cells and the FACS plot is not very convincing - many more events are required. Despite this, the plot shown indicates that there might even be two populations, one CD19^{high} and the other CD19⁺, which would be possible to determine upon acquiring many more events. Likely also affect the results.

19. What is the variation in proportion of Tbet+ cells in patients with SLE?

20. Based on the gated cells, a bimodal pattern is observed in SLE as in HDs except that the ratio between positive and negative cells is different, with the proportion of potentially positive cells being very low in SLE. Using the MFI rather than gating on positive cells will affect the results if the ratios between positive and negative cells vary between the different samples.

21. A negative control for A2aR expression was not shown to demonstrate if Tbet- B cells were truly negative for A2aR expression - is A2aR expressed on all B cells with Tbet+ B cells expression enhanced?

22. Are the MFIs between healthy and SLE patients comparable (run together or with laser standardisation using application settings)? If so, direct comparison would be preferable (changes in measured staining intensity may also be reflected in the internal MFI fold change).

23. The difference between HD and SLE based on these data are statistically significant but not convincing, not with so few Tbet+ cells to start with, not with the variation between samples, not with the bimodal pattern and not with the low A2aR expression levels.

24. Do human Tbet^{int} B cells express any A2aR - why has the focus been only on Tbet^{hi}?

25. Whether A2aR expression is restricted to only Tbet+ B cells is unclear, or to other cell types is also unclear

26. The manuscript lacks demonstration of the capacity of A2aR agonists to directly act upon human Tbethi B cells. It would be valuable to test in vitro whether human Tbethi B cells were responsive to A2aR agonism (any signal activation upon treatment) or whether A2aR agonists could induce apoptosis in human Tbet+ B cells.

Minor comments:

- Figure 3c: Scale bar not clear for histology images and missing details of magnification.
- Flow cytometry methods are not clear in which antibodies were employed for human studies and murine studies – a table would possibly be easier to discern clearly.
- How were cells from murine liver prepared for flow cytometry?
- Not clear if samples were frozen or prepared for flow cytometry immediately.
- Not clear if human samples were acquired on the cytometer at the same time or using standardised application settings in order to have robust MFI comparison across samples.
- Histology methods lack detail of size of sections. Was scoring conducted blind and how many scorers used?

We thank the reviewers for their careful reading and constructive criticism of our manuscript. We have addressed all of the concerns raised, and have incorporated suggested changes and discussion in the revised manuscript, which is much improved. We believe that the revised manuscript describes important findings that are relevant to both our understanding of the fundamental biology of T-bet+ B cells, and eventually to human health and the treatment of autoimmunity.

Major changes to the manuscript

Due to restrictions from the ongoing Covid-19 pandemic, shortcomings with current methods to detect the A_{2A} receptor at the protein level, and input from the reviewers, we have decided to omit the flow cytometry staining studies the human A_{2A} receptor (i.e., Figure 4 in the original manuscript). However, to demonstrate the clinical potential of A_{2A} receptor stimulation in the treatment of SLE, we have replaced this figure with new data demonstrating that A_{2A} receptor agonism can still ameliorate certain features of disease in MRL/lpr mice after disease onset. This finding is significant as it validates not just the prophylactic effectiveness of A_{2A} receptor stimulation, but also its therapeutic effectiveness, which is the more clinically relevant scenario for the treatment of SLE in humans. While we are as yet unable to unequivocally demonstrate expression of the A_{2A} receptor in humans, we clearly demonstrate the clinical potential of A_{2A} receptor agonism in the treatment of SLE. We believe these changes significantly strengthen the manuscript.

Reviewer #1 (Remarks to the Author):

This study by Levack et al. focuses on an important therapeutic area, namely the pharmacological targeting of “age/autoimmunity associated B cells (ABCs)” in systemic autoimmunity. As the authors note, multiple studies over the past decade have reported that CD11c+Tbet+ B cells are increased in subjects with autoimmunity, that these cells likely contribute to disease manifestations, and that have defined the immune signals required for ABC formation. The overall model emerging from these studies (including prior work from the Winslow group) is that ABCs are a subset of memory B cells that can rapidly differentiate into plasma cells following secondary challenge. However, there are currently no effective strategies to deplete ABCs. Thus, it has been difficult to determine the relative contribution of ABCs (as opposed to other immune subsets) to the clinical manifestations of SLE. For this reason, the current study holds both mechanistic and potential translational significance.

First, the authors leverage published RNA-Seq data showing that A2aR expression is increased in E. muris infected CD11c+Tbet+ B cells. Next, they show that separate A2aR agonists deplete CD11c+Tbet+ B cells, via a mechanism likely related to induction of cell death, not cell migration or downregulation of CD11c/Tbet expression. Using Adora2a floxed mice, they confirm that A2aR agonists deplete CD11c+Tbet+ B cells in a B cell-

intrinsic manner. Next, the authors confirm that A2aR agonists are able to deplete CD11c+Tbet+ B cells in separate murine lupus models, resulting in ameliorated disease. Finally, the investigators show that T-bet-expressing B cells in both healthy controls and lupus patients express A2aR, suggesting that A2aR agonism may be an effective therapy in human SLE.

Overall, these experiments are logically designed, and the data appropriately interpreted, with relevant caveats acknowledged in the discussion. While these observations are important, it should be noted that the findings involving CGS-21680 treatment of MRL/lpr mice are largely consistent with an earlier published study (Zhang, et al. *Lupus*, 2011). Thus, the major new finding of this current manuscript is that A2aR agonism ablates CD11c+Tbet+ B cells. The authors propose that this mechanism likely accounts for the therapeutic benefits of A2aR agonism in SLE, although definitive proof is lacking because of known impacts of A2aR agonists on other immune lineages (Tfh, etc.). For this reason, greater mechanistic understanding of how A2aR agonism promotes CD11c+Tbet+ B cell depletion in contrast with impacts on other immune cells implicated in lupus pathogenesis would strengthen this study.

We thank the reviewer for their clear summary and helpful suggestions. In particular, we thank Reviewer #1 for their insightful comments, which we believe has significantly strengthened the manuscript. We agree that further study into how A_{2A} receptor agonism depletes CD11c+ Tbet+ B cells is an important avenue of research and have commented as such in the responses to the specific questions raised below.

Responses to concerns raised by Review #1

1. A2aR expression on murine B cell subsets: Are anti-murine A2aR antibodies available to measure surface A2aR expression on different B cell subsets following Ehrlichia infection and/or murine lupus models? The authors note that A2aR transcript is elevated on CD11c+Tbet+ B cells following E. muris infection, but don't confirm increased protein expression by flow cytometry in the murine studies (expression was only reported on human B cells). Given the distinct sensitivity of different B and T cell populations to A2aR agonists, it would be useful to compare A2aR expression on, for example, naïve B cells, CD11c+Tbet+ B cells, plasma cells, Tfh cells, Tregs, etc.

We are very interested in defining surface expression of the A_{2A} receptor on different cell subsets, but have been hampered by the lack of suitable flow cytometry reagents. The A_{2A} receptor is a G protein-coupled receptor, and as the majority of its epitopes are sequestered within the cell membrane, researchers have not been successful at generating an effective antibody to target the receptor. Because of this shortcoming, there are a limited number of anti-A_{2A} receptor antibody clones available for flow cytometry, and only one which has been reported to bind the receptor in mice. We hope to address these questions in future studies, once better reagents become available.

2. Although splenic CD11c+ B cells appear exquisitely sensitive to A_{2A}R agonists, it is surprising that T-bet+ B cells in lymph nodes, liver, BM are resistant to CGS-21680 (Fig. 1C). Do the authors have a better explanation for this? What is A_{2A}R surface expression on eYFP+ B cells at these different sites. Since T-bet is expressed by multiple B cell lineages (not just ABCs), is the phenotype of eYFP+ B cells different based on tissue location. Are all eYFP+ cells phenotypically ABCs (CD11b+CD11c+CD21 low, etc.)?

We found this observation interesting as well, and at this juncture can only provide hypotheses. Although it is formally possible that only spleen T-bet+ B cells express the A_{2A} receptor, we believe this is unlikely. Instead, there is mounting evidence in the literature that T-bet+ B cells found in different tissues are phenotypically different from those found in the spleen. A recent publication by Johnson et al. in 2020 (*Immunity* 52:726) demonstrated that influenza-specific splenic T-bet+ B cells expressed higher quantities of T-bet than those found in other tissues, and this difference may affect cell function. It is also possible that T-bet+ B cells found in the liver, lymph nodes and bone marrow do not express as much A_{2A} receptor as their splenic counterparts. Given the limitations in our ability to assess expression of the A_{2A} receptor in mice, as noted in response to comment #1, we are unable to address this hypothesis at this time.

Alternatively, it is possible that T-bet+ B cells found in different tissues are poised to respond differently to A_{2A} receptor signaling than T-bet+ B cells in the spleen, possibly owing to differences in external stimuli, or innate differences in the cells themselves. The exact mechanism by which A_{2A} receptor signaling induces cell death in splenic T-bet+ B cells and why this subset is more susceptible than other B cell subsets is also the subject of ongoing research. In response to the question, we have commented on this finding in the manuscript, as follows. lines 274-279 (in the marked copy of the revised manuscript)

“These data also indicate that splenic CD11c+ T-bet+ B cells are the primary targets of the agonist, perhaps due to differences in A_{2A} receptor expression between CD11c+ T-bet+ B cells residing at different anatomical sites, differences in external signals received by the B cells, or to inherent differences in the ability of CD11c+ T-bet+ B cells at various sites to respond to A_{2A} receptor stimulation.”

In response to the second part of the question, we have previously demonstrated that T-bet+ B cells present after day 30 post-*E. muris* infection are phenotypically ABCs (i.e., they are CD11b+ CD11c+ and CD21^{lo}; Yates et al., 2013). While we have not yet extensively characterized the eYFP+ cells generated in T-bet-Cre^{ERT2} x Rosa26^{eYFP} mice, our data support the conclusion that cells which expressed T-bet between days 15-25 (when the cells in our published studies were tamoxifen labelled), and persisted until day 37 post-infection, were either T-bet+ memory cells (ABCs) or ASCs. As the majority of eYFP+ B cells we detected were CD138-negative (see data below), the bulk of the population were likely T-bet+ memory cells, and perhaps a small percentage of GCs (Kenderes et al., 2018). In the study by Kenderes et al., we performed a similar labelling strategy that utilized AID-Cre^{ERT2} x Rosa26^{eYFP} mice and showed that the majority of the T-bet+ B cells were phenotypically equivalent to ABCs (a minor

portion of the cells were GCs and ASCs). Thus, we conclude that most of the eYFP cells we detected in the present studies were indeed ABCs. We have added a comment regarding this question in the manuscript, as follows.

Lines 264-268

“We have used a similar labelling technique using AID-cre^{ERT2} x Rosa26^{eYFP} mice, and have shown that tamoxifen-labelled eYFP+ cells present after day 30 post-infection are canonical ABCs. Likewise, the majority of eYFP+ cells from the T-bet-cre^{ERT2} x Rosa26^{eYFP} mice were CD138-negative, similar to T-bet+ B cells typically seen at this time point in wild-type E. muris-infected mice (Supplemental Figure S2A).”

Data demonstrating the percentage of CD138-negative cells among eYFP+ B cells are shown below.

E. muris-infected T-bet-cre ERT2 x Rosa26eYFP mice were fed tamoxifen chow from day 15 to day 25 post-infection, followed by administration of vehicle (n = 3, 6) or agonist (n = 4, 5) every other day for seven days, starting on day 30 post-infection; tissues were analyzed on day 37 post-infection. The dot plots demonstrate that 94% of the T-bet+ B cells were CD138-negative (i.e., not ASCs) in a representative vehicle-treated mouse.

3. MRL.lpr model:

- Autoantibodies: Since A2aR agonists predominantly deplete T-bet+ B cells, one would predict that IgG2a/c subclass autoantibodies would be most impacted by A2aR agonist treatment. Additional analysis of autoantibody subclasses would thus support the investigators model in which CGS-21680 treatment predominantly targets CD11c+Tbet+ ABCs.

We agree that given the well documented bias of T-bet+ B cells to undergo class-switching to IgG2a/c, this isotype is predicted to be the most likely to be affected by the loss of CD11c+ T-bet+ B cells. In response to the reviewer's suggestion, we analyzed the sera from CGS-21680-treated MRL mice, and observed a significant decrease in IgG2a anti-dsDNA antibodies in treated mice, as shown below. However, we did not detect a statistically-significant change in IgG2a anti-RNA or IgG2a anti-Smith antibodies. It is unclear why an effect was only observed in the anti-dsDNA antibodies, so this will be addressed in ongoing studies in greater depth. Our data suggest, nonetheless, that IgG2a production was affected by agonist treatment, most likely a consequence of the depletion of T-bet+ B cells.

Sera from the mice in figure 2A were analyzed for IgG2a anti-RNA, anti-dsDNA, and anti-Smith antibodies by ELISA. The graphs show relative absorbance values for the analyzed sera. Statistical significance was determined using a two-tailed un-paired t test or a Mann-Whitney test (middle plot).

We have commented on these data, as follows, and have included them in the Supplementary data.

Lines 340-345

“Anti-RNA and anti-dsDNA antibodies were reduced in MRL/lpr mice treated with CGS-21680, relative to vehicle-treated control mice, while anti-Smith antibodies were not significantly changed (Figure 3A). IgG2a-specific anti-dsDNA were similarly reduced, although anti-RNA and anti-Smith IgG2a were unchanged (Supplemental Figure S5A). The reduction of IgG2a anti-dsDNA is consistent with a decrease in T-bet+ B cells, which characteristically undergo IgG2a switching⁵⁶.”

- Proteinuria: Despite statistically significant histologic decrease in glomerulonephritis, no difference was observed in degree of proteinuria. These appear predominantly driven by an absence of worsening proteinuria in the vehicle-treated animals, and contrasts with the findings of Zhang, et al. (Lupus, 2011). Do the authors have an explanation for this, since proteinuria is typically observed in the MRL.lpr model at this age. Were any other markers of renal dysfunction assessed (BUN, creatinine)?

One possible explanation for the differences between our data and those reported by Zhang et al., is that mice from the Zhang study were treated with CGS-21680 daily, albeit at a lower dose, whereas mice in our cohort were treated twice weekly. We chose our regimen as we believed it to be more clinically relevant, and less stressful to the mice. Additionally, Zhang et al., used a different, more sensitive method of protein quantification that was not available to us. We have made note of this in the manuscript. Unfortunately, we were unable to employ additional assays for renal dysfunction.

Lines 351-355

“In contrast with previous studies, we did not observe a significant change in proteinuria in mice treated with CGS-21680 compared to controls ($p = 0.1694$) (Supplemental Figure S5B).”

- Mortality curve: Given the striking reduction in splenomegaly/lymphoid hyperplasia, and decrease in activated B and T cells following CGS-21680 treatment, it is surprising that all the treated animals succumbed from autoimmunity with a modest

(non-significant) extension in median survival. Do the authors have an explanation for this? Could this be related to adenosine receptors desensitization following repeated agonist stimulation (which may limit human translational potential).

It is possible that overall mortality was unaffected due to A_{2A} receptor-induced tolerance in the mice which reduced the efficacy of the drug treatment. Another explanation is that in the polygenic MRL model many different cells contribute to disease, not all targetable by the agonist. Indeed, it is difficult to completely ameliorate disease in MRL/lpr mice, and many unrelated factors may contribute to mortality. We have modified the manuscript to reflect these possibilities, as follows.

Lines 355-362

*“CGS-21680 treatment modestly improved survival time among MRL/lpr mice, with a median survival time of 21.8 weeks among vehicle-treated mice, compared to a median survival time of 32.1 weeks among agonist-treated mice, although this difference was not statistically significant ($p = 0.535$), likely owing to the inherent variability among MRL/lpr mice, the contribution of other A_{2A} receptor tolerant cell types to disease, or to A_{2A} receptor desensitization following repeated agonist treatment (**Supplemental Figure S5C**).”*

Reviewer #2 (Remarks to the Author):

This original research manuscript by Levack, Winslow and colleagues reports for the first time, to the best of my knowledge, that use of adenosine 2a receptor (A2aR) agonists depletes CD11c+ T-bet+ B cells in ehrlichial-infected and lupus-prone mice. CD11c+ T-bet+ B cells have emerging roles in maintaining inflammation during infection and autoimmunity, however the role of A2aR activation in these B cells is undefined (and even for other B cell subsets remains largely undefined). Importantly, this manuscript demonstrates that A2aR activation reduces histological and biochemical features of lupus in mice, although not clinical disease per se. Finally, this report confirms previous reports in mice, that human CD11c+ T-bet+ B cells express relatively high A2aR compared to other B cells. Collectively, this study represents a highly novel and potentially important contribution to these fields and the broader immunology research community, but a number of issues need to be addressed before acceptance can be considered.

We thank the reviewer for their helpful suggestions. We would like to particularly thank Reviewer #2 for their careful reading of the manuscript, which has allowed us to improve the clarity of the presented work. We agree that the research presented herein has significant potential for multiple fields of study. We have addressed all of the Reviewer's concerns and in particular have performed the *in vitro* studies they have requested.

Responses to concerns raised by Reviewer #2

1. The Abstract claims to report the impact of A2aR activation on CD11c+ T-bet+ B cells however review of the data shows that either CD11c+ OR T-bet+ B cells (not CD11c+ T-bet+ B cells) were studied, presumably because they can be largely used interchangeably. The authors should comment on this in the manuscript.

CD11c and T-bet were used interchangeably in the manuscript, as we have shown that during *E. muris* infection, all CD11c+ B cells are T-bet^{hi} and, vice versa, all T-bet^{hi} B cells are CD11c-positive. The reference to these studies was unintentionally omitted and has been included in the revised manuscript (see below). However, as we had not formally demonstrated that all CD11c+ B cells are T-bet^{hi} and vice versa in the MRL/lpr model, we have modified our gating strategy to specifically interrogate CD11c and T-bet double positive B cells, as shown below. Indeed, the effects of agonist treatment are more pronounced in this analysis. We have modified Figure 2 to reflect this additional gating strategy, as shown below.

Lines 234-236 (in the marked copy of the revised manuscript)

"CD11c+ B cells and T-bet^{hi} B cells are referred to interchangeably as we have shown that these cells represent the same population at this time point in E. muris infection⁵."

Figure 2a. The zebra plots and graphs show the percentages and numbers of CD11c+ CD19+ cells and T-bet+ CD19+ B cells from vehicle and CGS-21680-treated MRL/lpr mice. Statistical significance was determined using two-tailed un-paired t tests.

2. Flow cytometry: The manuscript does not provide the full gating strategies used to study key murine and human immune cell populations (and as prescribed in the “Reporting Summary”).

The full gating strategies have been included in the supplemental data. All cells analyzed by flow cytometry were pre-gated on singlets and lymphocytes based on forward and side scatter, as shown below.

Supplemental Figure S1A. All cells analyzed by flow cytometry were pre-gated on singlets and lymphocytes based on forward and side scatter analysis, as shown in the contour plots.

3. Statistical analysis: There is no comment of normality of data, which should be provided (and as prescribed in the “Reporting Summary”).

Normality tests have been performed and the statistical tests have been modified, where appropriate.

4. Although this study uses the selective A2aR agonist Regadenosen and A2aR KO mice to confirm some findings, the study largely relies on the use of the A2aR agonist CGS-21680, which can also bind A1R and A3R albeit to a lesser extent (doi: 10.1007/s11302-015-9460-9). This reference and caveat should be noted in the manuscript.

It is true that CGS-21680 can bind the A₁ and A₃ receptors as well as the A_{2A} receptor,

although with much lower affinities (see Alnouri et al., *Purinergic Signal* 11:389). However, using Mb1-cre Adora2a^{flox/flox} mice we show that CGS-21680 acts on T-bet+ B cells via the A_{2A} receptor, as we did not observe any effects in the targeted mice, which still express the A₁ and A₃ receptors. Further research is needed to address this question in MRL/lpr mice, but the current data strongly suggests that our finding that the A_{2A} receptor is the major, if not only, receptor targeted in this model. We have updated the manuscript to include this caveat.

Lines 439-444

“CGS-21680 was chosen for our studies as this drug is a well-documented and selective agonist for the A_{2A} receptor, although it has been reported to interact with the A₁ receptor and A₃ receptor with lower affinity⁵⁸. Using Mb1-cre Adora2a^{flox/flox} mice, we showed that CGS-21680 acts on T-bet+ B cells via the A_{2A} receptor. Additional studies will be required to corroborate these results in the MRL/lpr mouse model, although our data strongly suggest that these effects are mediated solely by the A_{2A} receptor in those mice as well.”

5. Based on comparisons of lymphoid tissues and liver, lines 230-231 claim “that splenic CD11c+ T-bet+ B cells are the primary targets of the [CGS-21680]” however the authors provide no explanation as to why. The authors should comment on this in the manuscript.

There is mounting evidence in the literature that T-bet+ B cells found in different tissues are phenotypically different from those found in the spleen. A recent publication by Johnson et al. in 2020 (*Immunity* 52:726) demonstrated that influenza-specific splenic T-bet+ B cells expressed higher quantities of T-bet than those found in other tissues, and this difference may affect cell function. It is also possible that T-bet+ B cells found in the liver, lymph nodes and bone marrow do not express as much A_{2A} receptor as their splenic counterparts. Given the limitations in our ability to assess expression of the A_{2A} receptor in mice, due to the lack of reliable A_{2A} receptor-specific antibodies, we are unable to address this hypothesis at this time.

Alternatively, it is possible that T-bet+ B cells found in different tissues are poised to respond differently to A_{2A} receptor signaling than T-bet+ B cells in the spleen, possibly owing to differences in external stimuli, or innate differences in the cells themselves. The exact mechanism by which A_{2A} receptor signaling induces cell death in splenic T-bet+ B cells and why this subset is more susceptible than other B cell subsets is also the subject of ongoing research.

In response to the question, we have commented on this finding in the manuscript, as follows.

lines 274-279

“These data also indicate that splenic CD11c+ T-bet+ B cells are the primary targets of the agonist, perhaps due to differences in A_{2A} receptor expression between CD11c+ T-bet+ B cells residing at different anatomical sites, differences in external signals received by the B cells, or to inherent differences in the ability of CD11c+ T-bet+ B cells at various sites to respond to A_{2A} receptor stimulation.”

6. Although the Discussion outlines future studies, has the impact of CGS-21680 on murine and human CD11c+ T-bet+ B cells in mediating cell death in vitro been studied? Given the in vivo effects, even with a single dose (Fig 1B), it would appear relatively straight forward to incubate murine splenocytes and human PBMCs in vitro in the absence and presence of CGS-21680 and measure cell death in these and other cell subsets, and include this data in the manuscript.

In response to the reviewer's comment, we have conducted preliminary experiments (see data below). We purified PBMCs from both healthy donors and SLE patients and cultured them *in vitro* with or without CGS-21680 at 1 μ M for 48 hours. CD11c+ B cells from some patients were depleted following in vitro culture with CGS-21680, although the effects were not statistically significant on the group. These preliminary data suggest that A_{2A} receptor agonist treatment will be effective in humans. However, we also observed considerable heterogeneity within SLE patient samples (as would be expected from a heterogeneous disease like SLE) as CD11c+ B cells from two patients were depleted following A_{2A} receptor stimulation while 3 were not. Once again, given the constraints due to covid, we were unable to collect sufficient patient samples to be able to effectively stratify the patients and to determine why certain donors responded to A_{2A} receptor stimulation and why others did not. It is possible that CD11c+ T-bet+ B cells from different SLE patients express different quantities of A_{2A} receptor or that differences in disease activity or effects from other SLE treatments alter the effectiveness of A_{2A} receptor-mediated depletion. This finding suggests that the therapeutic use of A_{2A} receptor for the treatment of SLE will need to be tailored to those who are most likely to respond. Because we were unable to fully resolve this question, we have not included these data in the revised manuscript. We plan to continue this approach in ongoing studies.

PBMCs were collected from SLE donors (n = 5) and in vitro cultured for 48 hours with 1 μ M CGS-21680 or vehicle control. Flow plots show the percentage of CD11c+ B cells from a patient that responded to treatment after in vitro culture. Data were analyzed using a two-tailed paired t test (p = 0.2313).

Minor comments

1. The manuscript should adopt IUPHAR/BPS nomenclature for adenosine receptors (<https://www.guidetopharmacology.org/>).
2. Line 45: The capitalization of “Influenza” and “Hepatitis” appears unnecessary.

3. Line 49: Correct on first use “Ehrlichia muris” to “E. muris”.

4. Lines 132-137: In addition to amount of drug per mouse, the Methods should also provide the approximate doses (approximate amounts/kg) for transparency and comparison to other studies. Further this text should clarify that these does refer to both the infection and lupus models.

The manuscript has been updated with the requested changes.

5. CGS-21680 can reduce weight in other mouse models of disease to confound results (doi: 10.1016/j.intimp.2019.04.037 and 10.1007/s11064-019-02745-0). The authors should comment in the manuscript whether the CGS-21680 regime used in the present study also impacted weight.

Weight was not assessed in these studies. However, as CGS-21680 treatment affected disease burden, change in the weight of MRL/lpr mice could be due to direct effects of CGS-21680 treatment or as a result of changes in disease burden in the mice. As such, other assays such as histology and ELISAs would allow for quantitation of disease independently of these effects. We have commented on this caveat in the manuscript as follows.

Lines 362-363

“Weight was not assessed as a measure of disease outcome as CGS-21680 is known to affect weight gain in mice via an unrelated mechanism⁵⁷.”

6. Lines 139-161 and 163-169: The flow cytometry methodology is a little muddled by listing the anti-human antibodies amongst the anti-murine antibodies even though lines 139-161 only deals with murine spleen and lymph node cells to this point. These sections should be revised to improve clarity, perhaps by listing anti-human antibodies including the antiA2AR antibody (which seems to be the same for both mouse and human studies) at lines 163-169.

7. Line 174: correct “4%PFA” to “4% PFA”.

8. Lines 296 and 299: Although in Fig 3E legend, it would be helpful if the P vales for these non-significant differences (Fig 3D and 3E) were stated in the text also.

9. Figure 3 panel: Time points of tissues analyses should be stated for 3A, 3B and 3C. Also add the P value in panel for Fig 3D as for 3E.

10. Supplementary data should be proof-read/grammar checked for occasional typographical errors (e.g. “Bone Marrow” instead of “Bone marrow” in Fig S1D).

The requested changes have been implemented.

Reviewer #3 (Remarks to the Author):

Summary

Overall Aim

This study aims to identify a selective and specific method of targeting Tbet+ CD11c+ B cells, a potentially pathogenic B cell population in SLE (and other autoimmune conditions). In doing so this could provide a new pharmacological target for the treatment of SLE that results in specific depletion of B cells that drive pathology, without depletion of the entire B cell compartment.

Earlier findings

Prior work from this group identified an enhanced expression of adenosine receptor 2a (A2aR) in Tbet+ CD11c+ B cells, distinguishing this subset from other B cell populations in mice.

Aim

In light of their previous finding, the authors sought to selectively deplete established Tbet+ CD11c+ B cells using A2aR agonists. Two different immunological mouse models known to involve Tbet+ B cell responses were chosen for A2aR agonist mediated ablation; Ehrlichia muris intracellular bacterial infection, and systemic lupus erythematosus (lupus).

Current Findings

Daily treatment of E.muris infected mice with A2aR agonist CGS-21680 resulted in significant depletion (75-80%) of CD19+ CD11c+ B cells and CD19+ Tbet+ B cells in the spleen. One dose of agonist was sufficient to deplete (80%) of the CD11c+ B cells (24hrs post administration). A conditional A2aR KOs using mb-1 cre abolished depletion after treatment with the agonist whereas this was not the case with a CD4 cre, supporting that this was the result of direct action of the A2aR agonist on B cells and not via Th cells.

Similar depletion of Tbet+ and CD11c+ B cells was found in Lupus prone mice, MRL/lpr, in response to long term A2aR agonist. However, this treatment regime also resulted in depleted plasma cells (PCs) and plasmablasts (PBs) in the spleen and LNs. In this instance Tfh cells were also depleted which likely contributes to limited PC/PB. The A2aR agonist treatment in lupus prone mice broadly targeted multiple lymphocytes. Long term treated mice experienced lower autoantibody titres, RNA and dsDNA but not Smith antigen, reduced splenomegaly and lymphadenopathy, and improved kidney pathology with fewer infiltrating lymphocytes whereas proteinuria and overall-survival were not altered.

Finally, the authors investigated whether Tbet+ B cells in PB from healthy donors (HD) and patients with SLE express A2aR. The authors conclude that they do and that the levels are lower in the latter.

General comments

- The study adheres to ethical standards including ethics committee approval and consent procedures. Ethical approval and written consent for human PB studies listed and animal welfare standards adhered to.
- Standard biosecurity and institutional safety procedures have been adhered to

Strengths

- The aims of this study are robust, clear, and have clinical relevance
- Well written and clear
- Good summary of the literature
- Clear reasoning for the need and therapeutic potential specific Tbet+ B cell targeted treatment.
- Valuable application of a target already pharmacologically approved.
- E.muris study strong in determining Tbet CD11c B cell depletion as direct action of the agonist on the B cells. Well explored possibility of alternate migration and impact on other lymphocytes.

Limitations

- Only N=5 for most murine studies.
- Mouse models only run once
- Only N=6 for the human studies. The HDs were 25 to 55 years whereas the SLE patients were 41-65 years. With this low number of samples in each group and with no correction for e.g., age, disease activity, etc it is not possible to draw any conclusions from the results presented

We thank the reviewer for their comments, especially for their careful attention to our experimental design, particularly with regards to the human data collected in Figure 4. In response to their comments, we have made several major revisions, which we believe have significantly strengthened the manuscript.

Responses to concerns raised by Review #3

In response to input from reviewers, we have omitted the human flow cytometry studies in Figure 4, and have replaced it with new clinically relevant data. Consequently, we have omitted our responses to the concerns raised by the reviewer related to the data in Figure 4.

1. The elevated levels of A2aR expression in the bacterial-infection model relied on previous mRNA data. However, there are no data on protein levels, not on the CD11c+ / Tbet+ or other B cells, or other cell types in the mouse models. This is important as

mRNA and protein do not necessarily have to correlate. Moreover, evidence of A_{2A}R expression exclusively on Tbet+ B cells in the two models is lacking.

We are very interested in defining surface expression of the A_{2A} receptor on various subsets of cells, but have been hampered by the lack of suitable flow cytometry reagents. The A_{2A} receptor is a G protein-coupled receptor, and as the majority of its epitopes are sequestered within the cell membrane, researchers have not been successful at generating an effective antibody to target the receptor. Because of this shortcoming, there are a limited number of anti-A_{2A} receptor antibody clones available for flow cytometry, and only one which has been reported to bind the receptor in mice. We hope to address these questions in future studies, once better reagents become available. Regardless, we have concluded that CD11c+ Tbet+ B cells indeed expressed the receptor, as the cells responded to CGS-21680 treatment in wild-type mice, but failed to respond in Mb1-cre Adora2a^{flox/flox} mice, where all B cells lacked the A_{2A} receptor.

2. The development of Tbet+ CD11c+ B cells in the bacterial infection model is TLR7-independent whereas in the lupus model is TLR7-dependent. Hence, the Tbet+ CD11c+ B cells in the two models are not necessarily 'the same'.

We agree that there are likely some differences between CD11c+ Tbet+ B cells generated during *E. muris* infection and those generated during SLE, as there likely are between Tbet+ B cells generated in various other infections and disease. However, the field has overwhelmingly considered Tbet+ B cells (i.e., ABCs) to be a discrete B cells subset with multiple defining characteristics, including cell surface phenotype, transcription factor expression, antibody isotype utilization, and the conditions under which they develop. Previous work from our laboratory has extensively characterized *E. muris*-elicited CD11c+ Tbet+ B cells (Yates et al, 2013) and we have shown, by all available metrics, that they are very similar, if not identical to those generated during SLE.

While we were not able to confirm all the conclusions drawn from the *E. muris* model using the MRL/lpr model, owing to the lack of necessary gene-targeted MRL/lpr strains, these studies are the subject of ongoing research. We have commented on the point made by the reviewer, as follows.

Lines 301-304 (in the marked copy of the revised manuscript)

"Prior research in both mice and humans, has demonstrated that CD11c+ Tbet+ B cells are generated during SLE and that these SLE-derived CD11c+ Tbet+ B cells appear similar, if not identical, to CD11c+ Tbet+ B cells that arise during E. muris infection^{10,16,26}."

2. Are there any human chronic bacterial infection diseases where Tbet+ CD11c+ B cells accumulate? This would be more relevant than SLE and potentially very interesting.

We agree that the effects of A_{2A} receptor stimulation on chronic infection would be of significant interest and this question is the subject of future research. We have commented on this question in the manuscript, as follows.

Lines 429-436

“Our findings also suggest that A_{2A} receptor agonist-mediated depletion of CD11c+ T-bet+ B cells can alleviate disease burden in other illnesses where CD11c+ T-bet+ B cells have been implicated in disease pathogenesis, such as RA, MS, and Sjögren’s Syndrome^{14,15,54-57}. Indeed, previous work has demonstrated that CGS-21680 treatment reduced the autoantibody response and disease burden in a mouse model of RA⁴⁰. In addition, targeted depletion of CD11c+ T-bet+ B cells yield critical data in the study of other disease contexts such as during chronic viral and bacterial infections in humans.”

3. Treatment of *E.muris* infected mice with A2aR agonist CGS-21680 apparently depletes CD19+ CD11c+ B cells and CD19+ Tbet+ B cells in the spleen. However, whilst still being referred to as Tbet+ CD11c+ B cells, the authors do not co-stain Tbet and CD11c+ on the B cells which is important to clearly demonstrate that it is the same population

CD11c and T-bet were used interchangeably in the manuscript, as we have shown that during *E. muris* infection, all CD11c+ B cells are T-bet^{hi} and, vice versa, all T-bet^{hi} B cells are CD11c-positive. The reference to these studies was unintentionally omitted and has been included in the revised manuscript (see below). However, as we had not formally demonstrated that all CD11c+ B cells are T-bet^{hi} and vice versa in the MRL/lpr model, we have modified our gating strategy to specifically interrogate CD11c and T-bet double positive B cells, as shown below. Indeed, the effects of agonist treatment are more pronounced in this analysis. We have modified Figure 2 to reflect this additional gating strategy, as shown below.

Lines 234-236

“CD11c+ B cells and T-bet^{hi} B cells are referred to interchangeably as we have shown that these cells represent the same population at this time point in *E. muris* infection⁵.”

Figure 2a. The zebra plots and graphs show the percentages and numbers of CD11c+ CD19+ cells and Tbet+ CD19+ B cells from vehicle and CGS-21680-treated MRL/lpr mice. Statistical significance was determined using two-tailed un-paired t tests.

4. Cell numbers of other B-cell subsets are not shown, hence potential effects on these are not clear

While we agree that this is an interesting question, we did not evaluate the effects of A_{2A} receptor stimulation on other subsets of B cells, as the focus of the manuscript was on T-bet⁺ B cells. The time points analyzed were too late to observe significant populations of germinal center cells or plasmablasts; however, bone marrow and splenic ASCs did not appear to be affected by A_{2A} receptor stimulation (see data below). Moreover, the total number of B cells was unchanged in CGS-21680-treated mice, compared to vehicle controls, suggesting that A_{2A} receptor stimulation did not have pronounced effects on other populations of B cells (see the data below). We have included these findings as supplemental data, and have revised the manuscript as follows.

Lines 242-246

“While A_{2A} receptor stimulation depleted CD11c⁺ T-bet^{hi} B cells, CD11c-negative T-bet^{lo} B cells as well as ASCs and total B cells were largely unaffected, suggesting that these cells do not express, or express less A_{2A} receptor, possibly indicating a correlation between A_{2A} receptor expression and T-bet expression (Supplemental Figure S1C and D).”

E. muris-infected female C57BL6/J mice were treated with DMSO (vehicle; n = 7) or CGS-21680 (agonist; n = 6) every day for seven days starting on day 30 post-infection; splenocytes were analyzed by flow cytometry on day 37 post-infection. Statistical significance was determined using two-tailed un-paired t tests.

5. What impact does depletion of this cell type have on the infection clearance etc?

We assayed bacterial load in CGS-21680-treated mice and did not observe any significant changes in bacterial clearance. However, we did not expect the dosing regimen of CGS-21680 to affect bacterial clearance, as we have shown that long-term IgM, which we have shown is protective, is unaffected by T-bet⁺ B cell depletion. Thus, in the ehrlichia-infected mice, protection is afforded by multiple immune mechanisms. We have included a comment regarding the reviewer's concern in the revised manuscript, as shown below.

Lines 236-240

“CGS-21680 treatment eliminated nearly all the CD11c+ T-bet^{hi} B cells in *E. muris*-infected mice within seven days post-administration (**Figure 1A and Supplemental Figure S1A**). Bacterial load was unchanged in CGS-21680-treated mice, likely because multiple immune mechanisms (e.g., IgM) are protective in infected mice (**Supplemental Figure S1B**).”

Livers from vehicle- or CGS-21680-treated mice from figure 1A were analyzed for bacterial load by qPCR. The plot shows the number of bacterial copies for each mouse. Statistical significance was determined using a two-tailed unpaired t test.

6. Depletion of Tbet+ and CD11c+ B cells was also found in Lupus prone MRL/lpr mice. However, it is unclear whether the observed changes can be attributed to the A2aR agonist treatment depleting of Tbet+ B cells, as other cell types were also affected by the treatment, and it appears to be a more general immunosuppression in this disease setting (impact on plasma cells, Plasma blasts, and the T cells compartment).

7. It is no longer a targeted depletion. Therefore, interpretation of the follow up clinical pathology can no longer be attributed to Tbet+ B cells but could be the result of more general immunosuppression.

As noted in the Discussion section of the original manuscript (lines 404-419) there may be additional mechanisms whereby A_{2A} receptor stimulation ameliorates disease in lupus-prone mice. We have focused on the effects of A_{2A} receptor stimulation on Tbet+ B cells, as these cells are well-known to contribute to disease pathogenesis in lupus, and because these cells are clearly major targets of CGS-21680 treatment. Thus, we argue that our conclusion is valid and important. We are in the process of generating Adora2a flox mice on the MRL background; an approach that will allow us to resolve the contribution of Tbet+ B cell targeting to the protective effects of A_{2A} receptor stimulation. However, the studies are beyond the scope of the present study.

8. Anti-RNA and -DNA antibodies were reduced, however, these cannot be referred to as pathogenic, this was not tested and it is not known whether all e.g., anti-DNA antibodies are pathogenic

While autoantibodies are indicative of disease pathology, we agree that the exact roles of these antibodies in disease pathogenesis are not known. We have modified the manuscript accordingly.

9. Whether there is a general decrease in antibody levels and if ‘non-pathogenic’ antibody production is also impacted was not tested

Given the effects of CGS-21680 treatment on the number of ASCs, and on anti-dsDNA and anti-RNA antibodies, it is possible that treatment could affect total antibody levels. However, as we were focused on the effects of A_{2A} receptor stimulation on antibody levels associated with disease severity, total antibodies were not assessed.

10. Authors conclude that proteinuria levels have a trending reduction in agonist treated mice. However, these groups are always overlapping, and hence there is no reduction, and it is stretching it to refer to as a trend

While we respectfully disagree, we have moved the data to the supplement, and have modified our phrasing, as follows.

Lines 351-355

“In contrast with previous studies, we did not observe a significant change in proteinuria in mice treated with CGS-21680 compared to controls ($p = 0.1694$) (Supplemental Figure S5B).

11. The overall-survival after treatment is not statistically significant, there are far too few animals to observe a potentially significant effect, if there is any at all

As we stated in the manuscript, the differences were modest. We agree, however, that the limited numbers of mice available to us did limit our ability to assess significance. Because of this limitation, we have moved these data to the supplementary data, and have revised the manuscript, as follows.

Lines 355-362

“CGS-21680 treatment modestly improved survival time among MRL/lpr mice, with a median survival time of 21.8 weeks among vehicle-treated mice, compared to a median survival time of 32.1 weeks among agonist-treated mice, although this difference was not statistically significant ($p = 0.535$), likely owing to the inherent variability among MRL/lpr mice, the contribution of other A_{2A} receptor tolerant cell types to disease, or to A_{2A} receptor desensitization following repeated agonist treatment (Supplemental Figure S5C).”

12. Whether A2aR expression is limited to Tbet+ B cells in the Lupus prone mice was not analyzed. The referenced article which details A2aR enhanced expression is from *E.muris* infection. It would be necessary to explore A2aR expression on Tbet+ and Tbet- B cells in the lupus models, and on the T cells in these animals.

We are very interested in defining surface expression of the A_{2A} receptor on subsets of B cells, but have been hampered by the lack of suitable flow cytometry reagents. The A_{2A} receptor is a G protein-coupled receptor, and as the majority of its epitopes are sequestered within the cell membrane, researchers have not been successful at generating an effective antibody to target the receptor. Because of this shortcoming, there are a limited number of anti-A_{2A} receptor antibody clones available for flow cytometry, and only one which has been reported to bind the

receptor in mice. We hope to address these questions in future studies, once better reagents become available.

13. Another lupus model was also employed but with a different treatment regime. This makes it difficult to compare if the targeted effect of the A2aR agonist found in the short-term treatment was the result of the model itself or the reduced treatment time?

We agree that direct comparisons cannot be made between these experiments but nonetheless feel that an additional model provides more support for our observations that treatment can affect SLE. Even with a significantly reduced treatment regimen given after disease onset, we still see effects on T-bet+ B cells in the SLE123 mice, just as in the MRL/lpr mice. In response to the reviewer's concern, we have made additional note of this caveat in the manuscript, as follows.

Lines 326-331

*“CGS-21680 treatment significantly reduced the number of T-bet+ B cells in the spleens of nine-month-old SLE1.2.3 mice treated every other day for five days (**Supplemental Figures S3C**). This short-term treatment did not result in the reduction of ASCs, as was observed following long-term treatment of MRL/lpr mice, although differences in treatment regimens complicate additional comparisons (**Supplemental Figure S3D**).”*

Human HD and SLE patients

14. In the figure (Fig. 4), 2.22% of peripheral blood B cells express Tbet. This population is not very distinct and the variation between donors is not shown. Maybe it varies a lot, which would make it difficult to gate these cells and it would affect gating of the cells and detection of A2aR considerably.

15. The gated Tbet+ cells from HD demonstrate a bi-modal pattern of A2aR expression in Tbet+ cells and a lack of expression in Tbet- cells. However, in the absence of an FMO, it is unclear whether there is a lack of expression or just lower levels in the Tbet- cells.

16. As the proportion of Tbet+ cells is low, and since the proportion in the other samples is unknown, the variation could stem from very few Tbet+ cells, and affect the variation substantially.

17. Adding to this the bi-modal A2aR expression pattern, which might be reflected in the variation between samples in terms of A2aR MFIs, 3-fold on Tbet- and 6-fold on Tbet+ cells. Thus, despite a calculated statistically significant difference, the data are not convincing.

18. Performing the same experiment on PB from patients with SLE showed a very low proportion (<1%) of Tbet+ cells and the FACS plot is not very convincing - many more events are required. Despite this, the plot shown indicates that there might even be two populations, one CD19high and the other CD19+, which would be possible to determine upon acquiring many more events. Likely also affect the results.

19. What is the variation in proportion of Tbet+ cells in patients with SLE?

20. Based on the gated cells, a bimodal pattern is observed in SLE as in HDs except that the ratio between positive and negative cells is different, with the proportion of potentially positive cells being very low in SLE. Using the MFI rather than gating on positive cells will affect the results if the ratios between positive and negative cells vary between the different samples.

21. A negative control for A2aR expression was not shown to demonstrate if Tbet⁻ B cells were truly negative for A2aR expression - is A2aR expressed on all B cells with Tbet⁺ B cells expression enhanced?

22. Are the MFIs between healthy and SLE patients comparable (run together or with laser standardisation using application settings)? If so, direct comparison would be preferable (changes in measured staining intensity may also be reflected in the internal MFI fold change).

23. The difference between HD and SLE based on these data are statistically significant but not convincing, not with so few Tbet⁺ cells to start with, not with the variation between samples, not with the bimodal pattern and not with the low A2aR expression levels.

24. Do human Tbet^{int} B cells express any A2aR – why has the focus been only on Tbet^{hi}?

25. Whether A2aR expression is restricted to only Tbet⁺ B cells is unclear, or to other cell types is also unclear

26. The manuscript lacks demonstration of the capacity of A2aR agonists to directly act upon human Tbet^{hi} B cells. It would be valuable to test in vitro whether human Tbet^{hi} B cells were responsive to A2aR agonism (any signal activation upon treatment) or whether A2aR agonists could induce apoptosis in human Tbet⁺ B cells.

We have not commented on the concerns raised above because the relevant human data has been omitted from the revised manuscript.

Minor comments:

- **Figure 3c: Scale bar not clear for histology images and missing details of magnification.**

Figure 3c has been updated with the requested changes, as shown below.

Kidneys from the mice in figure 2A were H&E-stained and were scored blindly for interstitial nephritis and glomerulonephritis at 20 weeks of age. Representative sections are shown (top: 1x, bottom: 20x). Lymphocyte and plasma cell aggregates are shown in the cortical medullary junction (red stars) and in the renal pelvis (blue stars). Patent (red arrows) and occluded capillary loops (black arrows) are shown. The graphs show the pathology scores. Statistical significance was determined using an ordinary two-way ANOVA with Sidak's multiple comparisons test.

- **Flow cytometry methods are not clear in which antibodies were employed for human**

The human data has been omitted.

- **How were cells from murine liver prepared for flow cytometry?**

The requested information has been added to the methods section of the manuscript as seen below.

Lines 188-190

"Livers were perfused with PBS and disaggregated using a 70µm cell strainer (BD Falcon). Cells were gradient separated using 40% Percoll (Sigma Aldrich) and erythrocytes were removed by incubation with ACK lysis Buffer (Quality Biological Inc)."

- **Not clear if samples were frozen or prepared for flow cytometry immediately.**

Samples were analyzed immediately after harvest. The methods have been updated to reflect this methodology.

Lines 178-179

"All flow cytometry analyses were performed on freshly harvested cells."

- **Not clear if human samples were acquired on the cytometer at the same time or using standardised application settings in order to have robust MFI comparison across samples.**

The human data has been omitted.

- **Histology methods lack detail of size of sections. Was scoring conducted blind and how many scorers used?**

The histology methods have been updated as follows.

Lines 195-203

“Histology. *Kidneys were harvested from mice and fixed in 4% PFA for at least 48 hours at room temperature. Fixed kidneys were paraffin embedded, cut into 5µm sections, stained with hematoxylin and eosin (H&E), and pathology blindly scored by a single pathologist at Histowiz (Brooklyn, NY) or by a certified pathologist at the University of Pittsburgh Medical Center, using a previously described metric⁵². In short, glomerulonephritis was assessed using a scale from one to six based on mesangial cellularity and expansion, the presence of patent capillary loops, glomeruli size, karyorrhexis, crescent formation, and sclerosis. Interstitial nephritis was graded on a scale from one to four based on the prevalence of lymphocytes and plasma cell infiltrates in the perivascular area and/or in the interstitial space.”*

REVIEWER COMMENTS

Reviewer #1 (Remarks to the Author):

This revised manuscript by Levack et al. describes the pharmacological targeting of "age/autoimmunity associated B cells (ABCs)" in systemic autoimmunity using an Adenosine A2a receptor agonist.

My previous comments regarding this study focused predominantly on: 1) need to quantify levels of adenosine A2a receptor protein expression on murine B cell subsets; and 2) the discrepancy between robust ABC depletion in the Ehrlichia model and a more modest impact on disease severity in lupus-prone models.

The authors have largely addressed these concerns with the caveat that adequate flow cytometry reagents to identify adenosine A2a receptor protein are lacking. The study has been strengthened by the addition of new murine lupus-model data showing therapeutic benefit for CGS-21680 in mice treated after disease onset (i.e. treatment initiation at 12 weeks vs. 8 weeks in the MRL.lpr model).

Given lack of effective therapies to target ABCs, the revised manuscript is significant. Although the removal of human SLE data weakens the importance of these findings somewhat, I agree with the authors decision to eliminate these data since they represented a weakness in the original submission.

Major comments:

- 1) How were 8 week and 12 week ages chosen for initiation of treatment in the MRL.lpr model? It would be useful to show data confirming that 8 weeks is "pre-disease onset" and 12 weeks is "established disease", since these timepoints are relatively close together. At a minimum, it would be helpful to the reader to document that class-switched autoantibodies typically develop after 8 weeks (but before 12 weeks) to validate the conclusion that CGS-21680 is an effective treatment of established disease.
- 2) In this regard, what are autoantibody titers pre-treatment in the MRL.lpr model? Rather than CGS-21680 differentially impacting anti-DNA vs. anti-RNA titers, couldn't an alternate conclusion be that RNA autoantibodies derive from long-lived plasma cells not impacted by agonist treatment (i.e. formed prior to treatment initiation)?

Minor comments:

- 1) Check labeling of Supplemental Figure 4 in the main text (data in Suppl. Figure 4 labeled as Suppl. Figure 3).
- 2) Data in Supplemental Figure 4C could be moved to the main paper (i.e. replication of CD11c+Tbet+ B cell depletion in NZM model). Showing robust depletion of ABCs in an independent murine lupus model is important and validates the overall findings.

Reviewer #2 (Remarks to the Author):

The authors appear to have addressed my concerns but the submitted file does not match the line numbers in the rebuttal, which was made harder by not being able to locate the marked version online. Thus, the manuscript, cannot be fully appraised as submitted. Furthermore, references 59-62 do not seem to be cited and thus many of the references appear to be incorrectly assigned.

Reviewer #3 (Remarks to the Author):

The authors have revised the manuscript in a satisfactory manner. The results and conclusions are much easier to follow, and the message is now clear.

Reviewer #1 (Remarks to the Author):

This revised manuscript by Levack et al. describes the pharmacological targeting of “age/autoimmunity associated B cells (ABCs)” in systemic autoimmunity using an Adenosine A2a receptor agonist.

My previous comments regarding this study focused predominantly on: 1) need to quantify levels of adenosine A2a receptor protein expression on murine B cell subsets; and 2) the discrepancy between robust ABC depletion in the Ehrlichia model and a more modest impact on disease severity in lupus-prone models.

The authors have largely addressed these concerns with the caveat that adequate flow cytometry reagents to identify adenosine A2a receptor protein are lacking. The study has been strengthened by the addition of new murine lupus-model data showing therapeutic benefit for CGS-21680 in mice treated after disease onset (i.e. treatment initiation at 12 weeks vs. 8 weeks in the MRL.lpr model).

Given lack of effective therapies to target ABCs, the revised manuscript is significant. Although the removal of human SLE data weakens the importance of these findings somewhat, I agree with the authors decision to eliminate these data since they represented a weakness in the original submission.

Responses to concerns raised by Reviewer #1

Major Comments:

1) How were 8 week and 12 week ages chosen for initiation of treatment in the MRL.lpr model? It would be useful to show data confirming that 8 weeks is “pre-disease onset” and 12 weeks is “established disease”, since these timepoints are relatively close together. At a minimum, it would be helpful to the reader to document that class-switched autoantibodies typically develop after 8 weeks (but before 12 weeks) to validate the conclusion that CGS-21680 is an effective treatment of established disease.

The 8- and 12-week timepoints were chosen for the initiation of treatment based on previous studies. Liu et al., (2018; Nature Scientific Reports 8:1308) showed that anti-dsDNA and anti-nuclear antibodies can be detected at 8 weeks of age in MRL/lpr mice but are much increased by 12 weeks of age. Moreover, Tilstra et al., (2018; The Journal of Clinical Investigation 128:4884) have demonstrated that activated T cells are present in the kidneys of 11-week-old MRL/lpr mice, indicating ongoing disease. These studies reveal that disease onset occurs by 12 weeks of age in MRL/lpr mice indicating that agonist treatment can be effective therapeutically after disease onset.

Our original data demonstrated that agonist treatment was effective when administered early at 8 weeks. Although we indicated that this timepoint is prior to disease onset, the published studies mentioned above indicate that some ANAs can be detected at 8 weeks of age, albeit at low quantities. This observation does not affect our conclusion that early agonist treatment was effective. Nevertheless, as a point of clarification, we have reworded the manuscript in three places to indicate that 8 weeks of age is prior to or early following disease onset in MRL/lpr mice.

In the Introduction we state the following.

Lines 113-116

“Moreover, A_{2A} receptor agonist treatment of MRL/lpr mice significantly reduced disease severity, compared to vehicle-treated controls when administered either early or late relative to the onset of disease.”

In the Results we state the following.

Lines 350-353

“To assess the potential of A_{2A} receptor stimulation as a treatment for SLE, we next analyzed disease severity in MRL/lpr mice that had been treated with CGS-21680 after disease onset. MRL/lpr mice were treated with CGS-21680 from weeks 12 to 20 of age and were sacrificed on week 20. This timepoint was chosen based on previously published studies^{58,59}.”

Lines 359-360

“Similar to mice treated with CGS-21680 starting at 8 weeks of age, splenomegaly was reduced following delayed treatment as well (Figure 5B).”

2) In this regard, what are autoantibody titers pre-treatment in the MRL/lpr model? Rather than CGS-21680 differentially impacting anti-DNA vs. anti-RNA titers, couldn't an alternate conclusion be that RNA autoantibodies derive from long-lived plasma cells not impacted by agonist treatment (i.e. formed prior to treatment initiation)?

We thank the reviewer for this comment and reference to these findings. As stated in response to the first comment, ANAs can be detected at 8 weeks of age in MRL/lpr mice, although at low quantities (Liu et al., 2018; Nature Scientific Reports 8:1308). We agree that the reduction of anti-dsDNA antibodies, but not anti-RNA antibodies, may indicate that A_{2A} receptor stimulation preferentially targeted plasmablasts, and not long-lived plasma cells. We have commented on this possibility in the manuscript, as follows.

Lines 356-359

“The reduction of anti-dsDNA antibodies and not anti-RNA antibodies could indicate that A_{2A} receptor stimulation preferentially targeted plasmablasts and not long-lived plasma cells as these subsets are thought to be enriched for anti-dsDNA and anti-RNA reactive B cells respectively^{58,59}.”

Minor comments:

1) Check labeling of Supplemental Figure 4 in the main text (data in Suppl. Figure 4 labeled as Suppl. Figure 3).

We have corrected the error.

2) Data in Supplemental Figure 4C could be moved to the main paper (i.e. replication of CD11c+Tbet+ B cell depletion in NZM model). Showing robust depletion of ABCs in an independent murine lupus model is important and validates the overall findings.

We also thank the reviewer for this suggestion. We have revisited these data, and regated the flow cytometry plots, using the same gating strategy as in the revised version of

Figure 2a, and find the data from the SLE1.2.3 mice to be striking, and significant (see below). However, given the limited number of animals that were available to us at the time, we would prefer to keep these studies in the supplement.

Nine-month-old SLE1.2.3 mice were treated with vehicle (n = 3) or agonist (n = 4) every other day for 5 days and splenocytes analyzed. The zebra plots and graphs show the percentages of CD19+ CD11c+ T-bet+ B cells. Statistical significance was determined using a two-tailed un-paired t test.

Reviewer #2 (Remarks to the Author):

The authors appear to have addressed my concerns but the submitted file does not match the line numbers in the rebuttal, which was made harder by not being able to locate the marked version online. Thus, the manuscript, cannot be fully appraised as submitted.

Furthermore, references 59-62 do not seem to be cited and thus many of the references appear to be incorrectly assigned.

We apologize for the problem that the reviewer had in viewing the changes to the manuscript. We believe that this confusion was due to the "track changes" function in Microsoft word.

With regard to the references, the original citation indicated numbers 59, 60, 61, and 62 as part of a range of references between 58 and 63, so references 59 to 62 were included in that citation but were not stated explicitly. We believe that the text is correct in the revised manuscript.

Reviewer #3 (Remarks to the Author):

The authors have revised the manuscript in a satisfactory manner. The results and conclusions are much easier to follow, and the message is now clear.

We appreciate the reviewer's comments and careful reading of our manuscript.

REVIEWERS' COMMENTS

Reviewer #1 (Remarks to the Author):

The authors have addressed each of my prior queries. The results and conclusions are presented in a clear manner in the revised manuscript.

Reviewer #2 (Remarks to the Author):

The authors have addressed my final, minor concerns.